# Comparison of state-of-the-art error-correction coding for sequence-based DNA data storage

Andreas L. Gimpel [1], Alex Remschak[1], Wendelin J. Stark[1], Reinhard Heckel [2] & Robert N. Grass [1] ✉

Many codecs with different error-correction approaches have been implemented for DNA data storage to date. However, no studies have systematically benchmarked codec implementations to establish their current state-of-the-art. Here, we use in silico and in vitro experiments to compare the performance of six representative codecs from literature. In isolation, these codecs can tolerate error rates up to 14% and a sequence loss of 65%. Under realistic conditions, we further establish that storage densities as high as 117 EB g$^{-1}$ are feasible using existing codecs and current synthesis and sequencing technologies. Verifying our results experimentally, we demonstrate data storage at 43 EB g$^{-1}$ using synthesis by material deposition and 13 EB g$^{-1}$ using electrochemical synthesis, employing existing codecs from literature. Besides closing in on the physical limits of DNA data storage, this study thus demonstrates the maturity of error-correction coding, defines its current state-of-the-art, and establishes best practices for codec benchmarking.

Since the inception of DNA data storage, a major challenge has been defining the rules for reversibly converting between digital information and DNA sequences. This challenge, which falls within the realm of coding theory for DNA data storage, mainly involves the design and implementation of effective encoders and decoders ("codecs"). For this, early demonstrations of DNA data storage focused on source coding (i.e., compression) to efficiently encode the mostly text-based data used at the time[1], e.g., by run-length encoding[2] or Huffmann codes[3,4]. Then, with the advent of array-based DNA synthesis and next-generation sequencing, channel coding came into focus to enable error-free recovery of binary data and overcome the necessity for manual intervention during decoding[5–7]. Still today, the majority of codecs employ linear block codes such as Reed-Solomon[7–9], Fountain[10,11], or repetition[6,12] codes with an inner/outer code separation strategy for error correction (see Supplementary Table 2 for an overview)[1].

For sequence-based DNA data storage with Illumina sequencing, which has been used for the largest demonstrations of DNA data storage to date[9,13,14], the use of channel coding is necessitated by the peculiar challenges posed by this workflow. First, array-based DNA synthesis is limited to sequence lengths of only a few hundred nucleotides (nt), requiring data segmentation across many individual sequences in an oligonucleotide pool[13,15]. Secondly, all biochemical steps of the DNA data storage workflow (see Fig. 1a) introduce errors into sequences and affect their distribution, potentially leading to sequence loss[16,17]. Thus, codecs must simultaneously compensate for nucleotide errors and sequence dropout, motivating the widely-used code separation strategy with an inner and outer code[1,13]. Beyond these basic considerations however, the breadth of intended applications and available technologies for DNA data storage each present individual challenges and requirements. Accordingly, many codecs are designed to support specific applications' error profiles (e.g., photolithographic synthesis[18], aging-induced decay[19,20], or nanopore sequencing[9,21,22]) or fulfill specific sequence constraints (e.g., GC content[6,10], homopolymers[5,7,10], k-mer frequency[11,23], motifs[11], or free energy[24]).

Besides the logical redundancy introduced by codecs, the inherent presence of many sequence copies during biochemical processing

[1]Department of Chemistry and Applied Biosciences, ETH Zürich, Zürich, Switzerland. [2]TUM School of Computation, Information and Technology, Technical University of Munich, Munich, Germany. ✉e-mail: robert.grass@chem.ethz.ch

**a** End-to-end overview of the DNA data storage pipeline

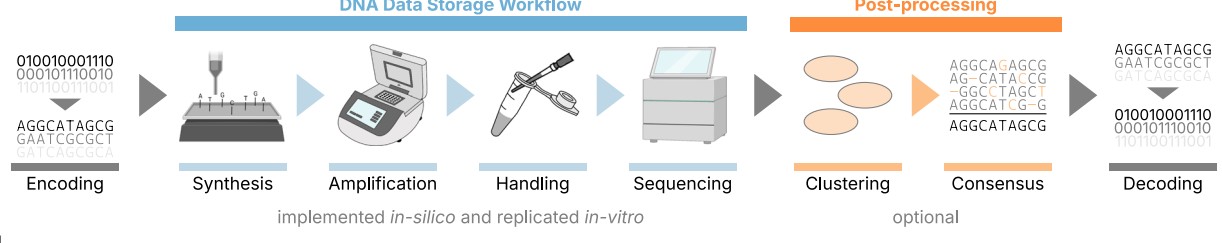

**Scope of this work:** end-to-end comparison of selected error-correction codecs across workflows

**b** Estimating error tolerance for successful decoding

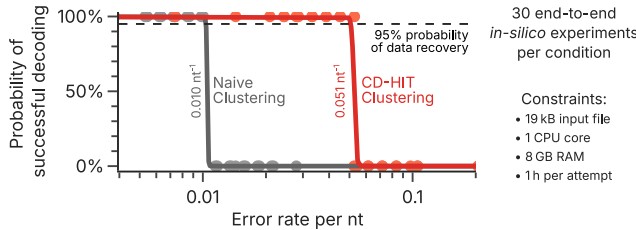

**c** Tolerances against sequence loss

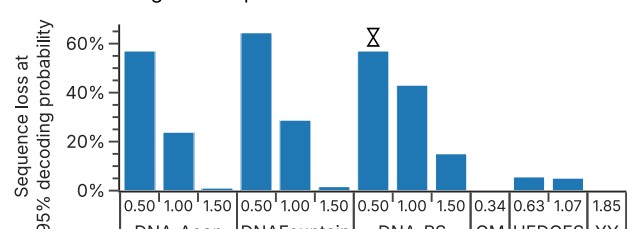

**d** Effect of clustering on error tolerances of codecs

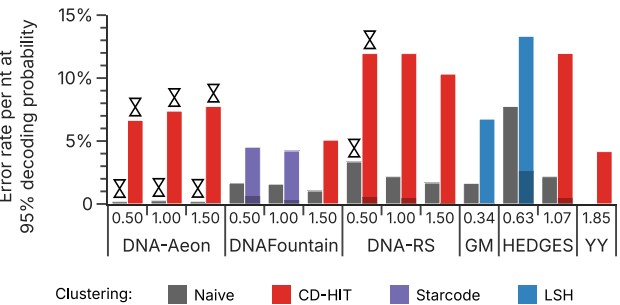

**e** Pareto fronts between errors and sequence dropout

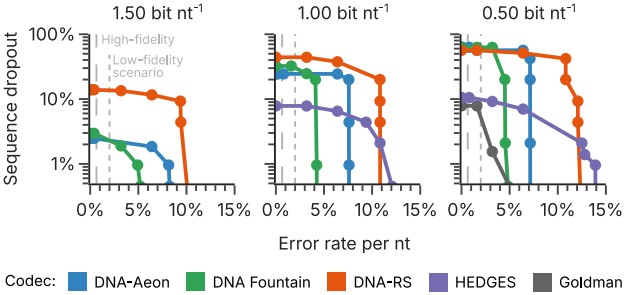

**Fig. 1 | Overview of the scope of this work and the evaluations of clustering algorithms and codecs. a** Overview of the data storage workflow considered in this work, including encoding of the data with a codec, an in silico or in vitro data storage workflow, post-processing by read clustering and generation of consensus sequences, and decoding of the data with a codec. **b** Exemplary outcome of a basic error scenario with naïve (gray) or CD-HIT clustering (red) upon variation of the overall error rate (see Methods). Individual points denote the outcomes of 30 individual iterations of the scenario. The solid lines represent the logistic regression performed to estimate the error rate at which data recovery succeeds with 95% probability (dashed gray line). The corresponding error rate is then used as performance metric in this work. **c** Tolerance of all codecs in the basic error scenario to sequence loss. All codecs use the best-performing clustering algorithm denoted in panel d. Conditions indicated with an hourglass (⧖) were limited by the time constraint (see Supplementary Figs. 6, 7). **d** Performance of each codec at its supported code rates in the basic error scenario, using both naïve clustering (gray bars) and the best-performing clustering algorithm for each codec (colored bars,

see Supplementary Table 3 for full data). The error rate shown corresponds to the combined rate of substitutions, deletions, and insertions at a fixed composition of 53% substitutions, 45% deletions, and 2% insertions, thereby resembling the experimental error pattern in ref. 17. Conditions indicated with an hourglass (⧖) were limited by the time constraint (see Supplementary Figs. 4, 5). Dark shading of the bars indicates the effective error rate remaining after the clustering step. **e** Pareto fronts of codec performance in a scenario combining errors at a fixed ratio of 53% substitutions, 45% deletions, and 2% insertions with sequence dropout. Feasible regions lie below the indicated pareto fronts, with the mean error rates of the high-fidelity (dashed line) and low-fidelity scenario (dotted line) indicated. The Yin-Yang codec, which did not have independent error-correction capabilities, was unable to decode the data in this scenario at all, given that neither it nor the clustering step was capable of compensating for any sequence dropout. Panel a) partially created in BioRender. Gimpel, A. (2025) https://BioRender.com/9wrsaez. Source data are provided as a Source Data file.

provides an additional layer of redundancy in DNA data storage, termed physical redundancy[13,16]. However, achieving DNA's extreme storage densities (theoretically up to 227 EB g$^{-1}$ for double-stranded DNA, see Supplementary Note 2) requires minimizing both logical and physical redundancy simultaneously, while maintaining sufficient redundancy to facilitate error-free decoding[25]. As a result, while logical redundancy is directly related to synthesis cost, only the product of logical and physical redundancy is relevant for the storage system's data density. Complicating matters further, sequencing also yields multiple reads per sequence, providing another source of redundancy through sequencing depth[13,16]. Evidently, there exist a trade-off between these forms of redundancy which is not reflected in the encoder's code rate (i.e., data bits stored per nucleotide). Thus, any

attempt at isolating the performance of a codec from these other sources of redundancy is futile without sufficient standardization of experimental conditions.

Nonetheless, codecs are still commonly evaluated across studies by simply comparing the level of logical redundancy used (i.e., the encoder's code rate), despite the often vastly different experimental conditions (e.g., synthesis provider, physical redundancy, sequencing depth)[11,24,26]. In recent years, codecs have also been increasingly compared through in silico simulations that artificially vary the error rate in their input data[11,23,24]. However, these comparisons lack an established baseline and the extent of their standardization (e.g., code rate, sequence length, file size) often remains unclear. As a notable exception, Ping et al.[24] were the first to evaluate two codecs in a standardized

**Table 1 | Codecs included in the in silico and in vitro benchmarks, as well as their properties**

| Name | Error correction | | Constraints | Clustering | Customizability | Code rates | Refs. |
|------|------------------|---|-------------|------------|-----------------|------------|-------|
| | inner | outer | | | | | |
| DNA-Aeon | Arithmetic | Fountain | GC, HP, motifs | CD-Hit | Broad | 1.50, 1.00, 0.50 | 11 |
| DNA Fountain | Reed-Solomon | Fountain | GC, HP | Naïve[a] | Broad | 1.50, 1.00, 0.50 | 10,27 |
| DNA-RS | Reed-Solomon | Reed-Solomon | None | None or LSH | Broad | 1.50, 1.00, 0.50 | 18,28,29 |
| Goldman | Parity | Repetition | HP | None | None | 0.34 | 6 |
| HEDGES | Convolutional | Reed-Solomon | GC, HP | None | Constrained | 1.07, 0.63 | 30 |
| Yin-Yang | None[b] | | GC, HP, ΔE | None | Only length[b] | 1.85 | 24 |

Clustering denotes the clustering approach used in the original study, if any. The pairing of codecs with clustering algorithms used in this study is detailed in Supplementary Table 3. Based on the benchmarking results (see below), only the DNA-Aeon and DNA-RS codecs were selected as the current state-of-the-art.
[a]Naïve clustering reduces sequencing data to its set of unique reads, i.e., removing duplicates without consensus generation.
[b]The Yin-Yang codec is a bit-to-base coding scheme, and does not include any error correction in itself.

experiment, using a serial dilution as a benchmark. Nonetheless, the absence of a generally accepted state-of-the-art and standardized benchmarks with experimental relevance currently impedes objective assessment of codec performance.

In this study, we systematically benchmark codecs for DNA data storage both in silico and in vitro to establish the current state-of-the-art. For this, six representative codecs selected from the literature in October 2023 (see Table 1) were tested in multiple standardized scenarios across a range of experimental conditions centered around the most common, sequence-based data storage workflow with Illumina sequencing[13,14]. For our in silico benchmarks, we first employ simple synthetic error scenarios (i.e., applying fixed error rates) commonly used in the literature. Then, we extend our analysis with comprehensive benchmarks considering the full spectrum of non-idealities present in real-world sequencing data (using DT4DDS[17]), and verify our results with standardized in vitro experiments. In doing so, we demonstrate the benefits of read clustering on codec performance, assess the transferability of synthetic benchmarks to realistic scenarios, and evaluate the limits of common experimental benchmarks. We establish and verify an experimental benchmark for codec performance, demonstrating the capabilities of existing codecs to achieve record-breaking storage densities. Our work presents an unbiased and standardized assessment of the current state-of-the-art in error-correction coding for DNA data storage, thereby providing both a suitable baseline and a framework for benchmarking for future studies.

## Results
### Selecting and standardizing codecs for benchmarking
For this study, we limited our benchmarking of codec performance to representative examples, using the availability of an open-source implementation with sufficient documentation, the presence of in vitro experiments in the original publication, the prominence in literature, and considerations for covering a broad range of approaches as selection criteria (see "Methods" and Supplementary Table 2, cut-off date October 2023). As a result, we selected DNA-Aeon by Welzel et al.[11], DNA Fountain by Erlich and Zielinski[10,27], DNA-RS by Heckel[18,28,29], an implementation of the codec used by Goldman et al.[6] ("Goldman"), HEDGES by Press et al.[30] and Yin-Yang by Ping et al.[24] (see Table 1 for an overview, and Supplementary Table 2 for a list of candidate codecs). All codecs were standardized to facilitate impartial performance comparisons by choosing their parameters—as far as possible—such that they yield similar code rates (0.50, 1.00, and 1.50 bit nt$^{-1}$) and sequence lengths (around 150 nt).

Importantly, codec parameters were not optimized specifically for this study. Instead, the parameters selected by the codecs' original authors were only adjusted to achieve standardized code rates and similar sequence lengths. This effectively changed only each codec's

overall redundancy level (via the code rate), whereas other codec features—such as the balance between inner and outer redundancy—were kept at approximately the level selected by the original authors. While this approach might lead to suboptimal codec configurations in our benchmarks, this approach mimics the use case for experimentalists and matches each codec's authors' selected implementation most closely. This was deemed important, as the codecs' original authors were assumed to have estimated their ideal parameters based on their specific use cases in DNA data storage. These selected parameters are listed in Supplementary Tables 5–9. Minor changes to some codec implementations were also required to facilitate automated testing and ensure representative performance (see Supplementary Note 1).

For an initial comparison, a basic error scenario was implemented, analogous to the most common synthetic performance benchmarks in the literature[11,23,24,30]. This scenario randomly introduces single-nucleotide errors to create 30 erroneous copies of each sequence generated by a codec, at a variable rate and specified composition (53% substitutions, 45% deletions, and 2% insertions, resembling the error pattern in ref. 17). Using this approach, a broad range of error rates were iteratively sampled in order to identify the error rate at which decoding started to fail (for a total of 30 decoding attempts per condition, see Fig. 1b and "Methods"). Then, as our standard metric of error tolerance, we report the error rate at which decoding still succeeded with 95% probability, based on a logistic regression of all 30 decoding outcomes (solid lines in Fig. 1b).

Notably, decoding was constrained to one hour, 8 GB of memory, and one CPU core per attempt, using an input file of 19 kB. The choice of time constraint and file size were based on limitations of the employed computing environment, and thus indirectly enforced a decoding speed of at least 5.4 bytes per second. While this minimum enforced decoding throughput was thus not based on practical considerations, it is nonetheless sufficiently permissive such that any codec which is limited by this constraint would be unsuitable for most applications of DNA data storage (e.g., cold storage of large archives). Accordingly, the time constraint did not limit decoding in the vast majority of cases, and we show the decoding performance relative to the runtime in Supplementary Figs. 4, 6, 9, 11, 12, and 13. Moreover, the constraint to one CPU core negated any undue disadvantage due to a lack of parallelization in a codec's implementation. However, this constraint thus overlooks inherent incompatibilities with parallel processing possibly present in some decoding algorithms, which represent a considerable disadvantage to their large-scale implementation.

Figure 1d shows the error tolerance of all codecs in this basic error scenario, without clustering and consensus generation by alignment (i.e., the set of all 30 erroneous reads per sequence are supplied to the

decoder directly, following the procedure by Erlich et al.[10]; shown as "Naïve", gray bars). Unsurprisingly, lower code rates (i.e., higher redundancy) led to increased error tolerances for all codecs. However, the time constraint limited decoding performance in several runs, especially for the DNA-Aeon codec (symbolized by $\bar{\times}$ in Fig. 1d, see also Supplementary Fig. 4). As a result, DNA-Aeon failed to decode the data within the time limit at error rates above 0.3%, irrespective of code rate. In contrast, the HEDGES codec, at 0.63 bit nt⁻¹, exhibited the highest error tolerance at 7.7%, more than double that of the runner-up, DNA-RS with 3.3% at 0.50 bit nt⁻¹. Thus, based only on this commonly used synthetic performance benchmark, the HEDGES codec would be considered the best-performing codec of our selection from the literature.

### Testing the benefits of read clustering on codecs' error tolerance

A codec's decoding capability is directly related to the balance between available redundancy and error frequency in the data. Contrary to most traditional coding channels however, the sequencing data used for decoding in DNA data storage is inherently replicated, i.e., multiple erroneous reads of each sequence are available. This inherent repetition code can be exploited by generating a less erroneous consensus sequence from the individual reads via clustering (see Fig. 1a). Therefore, read clustering promises to increase a codec's error tolerance without the need for additional logical redundancy, which is surprisingly seldomly exploited in the literature (see Table 1).

To assess the benefits of clustering on codec performance, we selected both established clustering algorithms from bioinformatics (CD-HIT, MMseqs2, Starcode)[31–33] as well as specialized DNA data storage clustering algorithms (LSH, Clover)[18,34] from the literature. First, parameters for all clustering algorithms were selected which maximized their sensitivity, accuracy, and specificity (see Methods and Supplementary Table 1). Then, each codec was paired with each clustering algorithm to assess compatibility and quantify improvements in error-correction capabilities. For this, the basic error scenario, i.e., the introduction of random errors into 30 sequence copies, was reused.

Comparing the previously established error tolerances without clustering ("Naïve", gray bars in Fig. 1d) to the best-performing clustering algorithm for each codec (colored bars in Fig. 1d) revealed that clustering and consensus generation improved codec performance in all cases. On average, tolerated error rates increased by 6.5 ± 2.5% in absolute terms, effectively more than doubling the error tolerance of most codecs. As expected, this positive effect on error tolerance was caused by a drastic reduction in the error rate within the clustered reads (as low as 0% up to an error rate of 5%, see Supplementary Figs. 1 and 2), thereby considerably reducing the effective error rate for the decoder. Our results match previous results on the error-correction capability of consensus generation[35], and thus highlight the benefits of exploiting the inherent redundancy in sequencing data. The latter is best illustrated by the performance of the Yin-Yang codec, an optimal bit-to-base coding scheme without error-correction capabilities[24]: it exhibited an error tolerance of 4.2% while relying solely on the indirect error correction conveyed by clustering. Specifically, the clustering step already completely eliminates all errors in the consensus reads up to this error rate (see also Supplementary Fig. 1), removing the need for any codec-level error correction for error-free data recovery. Thereby, its performance represents the baseline for every codec with additional error-correction capabilities in this basic error scenario.

Across all codecs, the tolerated error rate with clustering often exceeded 5%, well above the error rates commonly encountered after commercial synthesis and in common workflows[16,17,25]. Additional tests considering each error type in isolation (see Supplementary Figs. 8–10) also did not expose any major differences in error tolerance between error types. Notably, the performance of DNA-Aeon improved most drastically with clustering, achieving an error tolerance of 7.7% at

1.50 bit nt⁻¹. Evidently, the reduced workload conveyed by clustering lifted its limitation by the time constraint drastically (see Supplementary Fig. 4). This highlights another benefit to read clustering besides the exploitation of sequencing data's inherent redundancy: the acceleration of decoding pipelines through the reduction of codec workload by 1–2 orders of magnitude (depending on sequencing depth).

Interestingly, nine of the thirteen tested codecs and code rates performed best with the established general-purpose clustering algorithm CD-HIT (red bars in Fig. 1d). Unsurprisingly, this general preference for CD-HIT was caused by this clusterer's superior performance, yielding the fewest and least erroneous clusters across all tested clustering algorithms up to an overall error rate of around 10% (see Supplementary Figs. 1-3). In addition, several combinations of clustering algorithms and codecs failed completely (see Supplementary Table 3 for full results). Further investigation suggests incompatibilities exist between these clustering algorithms and the sequence features generated by some codecs (e.g., indexing regions, overlapping sections in the Goldman codec). Specifically, some codec-clustering pairings exacerbate sequence loss by clustering reads from different design sequences together. In some cases, up to 92% of sequences present in the sequencing data no longer appeared in the clustered reads, explaining the incompatibility of such codec-clustering pairings. Thus, the ideal clustering algorithms must reduce the error rate in consensus reads efficiently, thereby reducing the workload for the inner decoder, without causing sequence loss themselves, which drastically burdens codecs' outer decoders. Given the universal benefits of clustering identified above, the pairing between codecs and clustering algorithms established in Fig. 1d were used for all further in silico studies and in vitro experiments.

### Evaluating codecs' tolerance to errors and sequence dropout simultaneously

Errors within the DNA sequence are not the only type of fault occurring in the DNA data storage channel. Also sequence dropout, i.e., the absence of reads from some sequences in the sequencing data, requires correction by a codec[13,16]. However, codecs' tolerance to sequence dropout is rarely quantified in the literature, especially in combination with variable error rates. Thus, we extended the aforementioned basic error scenario with another variable, the fraction of sequences lost. As such, Fig. 1c shows the rate of sequence loss each codec tolerates in the absence of errors. The large differences in tolerated sequence loss between codecs and code rates—from 0% (Goldman and Yin-Yang) to 64% (DNA Fountain at 0.5 bit/nt)—highlight that codecs are not equally equipped to compensate for this unique problem in DNA data storage.

The addition of sequence dropout as a second error source also enabled the simultaneous quantification of codecs' tolerance towards errors and sequence dropout, uncovering any considerable tradeoffs. Figure 1e shows the resulting Pareto fronts, delimiting each codec's feasible region for the simultaneous correction of errors and sequence dropout.

In accordance with the previous results, lower code rates (i.e., higher redundancy) considerably extended each codec's feasible regions in Fig. 1e, by increasing their tolerances to both errors and sequence dropout. However, the extent to which each codec tolerated sequence dropout, even at minimal error rates, differed considerably. Especially HEDGES, previously identified as best-performing codec based on raw error-correction capability above, only tolerated up to 7.8% and 10.5% sequence dropout at 1.07 bit nt⁻¹ and 0.63 bit nt⁻¹, respectively. This improves only slightly on the 7.9% sequence dropout tolerated by the Goldman codec, with its basic repetition code at 0.34 bit nt⁻¹. In contrast, DNA-Fountain—including only a small RS code within each sequence to detect rather than correct an erroneous sequence[10]—previously exhibited a low error-correction capability (up to around 5% nt⁻¹) across all code rates. However, its tolerance to

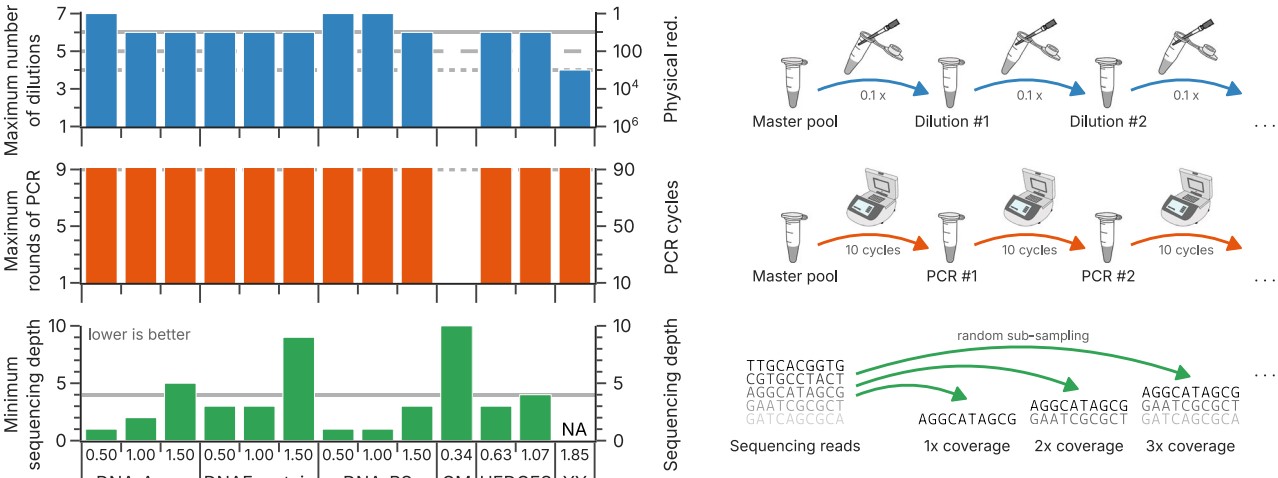

**Fig. 2 | Comparison with literature experiments replicated in silico.** All codecs were tested in three replicated literature experiments as reported by Erlich and Zielinski[10], Organick et al.[9,25], and Ping et al.[24]: serial dilution (blue), deep amplification (orange), and down-sampling (green). The individual workflows are illustrated on the right. In all cases, the best iteration at which all five repetitions succeeded are reported (see "Methods"). For comparison, the performance reported in the corresponding literature experiments by Organick et al.[9,25], (solid lines), Ping et al.[24] (dashed lines), and Erlich and Zielinski[10] (dotted lines) are shown. Source data are provided as a Source Data file.

sequence dropout increased drastically from 3% at 1.50 bit nt$^{-1}$ up to 63% at 0.50 bit nt$^{-1}$. Evidently, each codec's tolerances to errors and sequence dropout are balanced differently, such that neither metric in isolation can be considered as a robust performance benchmark.

The runtime constraint also limited codec performance of DNA-Aeon, DNA-RS, and HEDGES in this scenario (see Supplementary Fig. 11). Interestingly, both DNA-Aeon and HEDGES were only limited by time at high error rates while operating unconstrained at high sequence dropout, suggesting their implementations of within-sequence redundancy caused excessive runtimes. As previously, memory use did not limit any codec in this scenario.

In showcasing each codec's balance between correction of errors and compensation for sequence loss, Fig. 1e also highlights the benefits of a balanced approach to redundancy. The DNA-RS codec—and DNA-Aeon to a lesser extent—dominated the feasible regions of DNA Fountain and HEDGES, unless extreme sequence dropout (>50%) or error rates (>12%) occurred. Considering additionally that especially DNA-Aeon (at all code rates), but also DNA-RS (at 0.50 bit nt$^{-1}$), were partially limited by the time constraint (see Supplementary Fig. 11) at these extremes, these two codecs emerge as broadly superior in this synthetic analysis of errors and sequence dropout.

### Benchmarking codecs with common literature experiments

While the synthetic benchmarks used above clearly showcased each codec's theoretical capabilities under synthetic conditions, they are only a simplified representation of DNA data storage's true error channel. Despite the common use of these simple synthetic benchmarks for codec benchmarking in the literature, they overlook the non-ideal error patterns and biases present in experimental sequencing data—from error runs to skewed coverage distributions[16,17,36]. Thus, for all further benchmarks, we implemented realistic experimental workflows using the models implemented in the simulation software DT4DDS[17] (see "Methods"). This simulation software takes into account the aforementioned non-idealities which are missing from the synthetic benchmarks previously performed in the literature.

As a first realistic benchmark, three common experiments used by Erlich and Zielinski[10], Organick et al.[9,25], and Ping et al.[24] were replicated in silico. The three chosen scenarios covered the most common experiments in DNA data storage: a serial dilution to assess maximum storage density, a serial amplification to demonstrate copyability, and read down-sampling to quantify minimum sequencing depth (see Fig. 2).

In accordance with the original studies' results (gray lines in Fig. 2), our standardized, in silico replications of these literature benchmarks showcased the possibility of data recovery at high storage density and after deep replication. However, the serial dilution and serial amplification benchmarks exposed only minor performance differences between codecs at best (blue and orange bars in Fig. 2). In the serial dilution benchmark, the iterative 10-fold dilutions only partially resolved any differences between code rates of the same codec. This is expected, given that this benchmark uses oligo pools synthesized by material deposition (i.e., Twist Bioscience), whose high homogeneity and low error rate pose few challenges to codecs even at physical redundancies as low as 10×[25]. In the serial amplification benchmark, deep replication of these oligo pools by a high-fidelity (i.e., low-error) polymerase thus expectedly also did not lead to any discernable difference between codecs. Evidently, all codec implementations – with the exception of the Goldman codec—are similarly performant in these two literature benchmarks under equal and controlled conditions. This renders these two literature benchmarks ill-suited for comparing codec performance experimentally.

In contrast to the other two literature benchmarks, the down-sampling of sequencing data did expose considerable differences between codecs (green bars in Fig. 2). Given the low error rates associated with synthesis by material deposition[17], this benchmark relies mostly on tolerance to sequence dropout rather than error correction, thereby it is expected to favor codecs such as DNA Fountain. However, both DNA-Aeon and DNA-RS outperform DNA Fountain considerably across all code rates, requiring as few as only one sequencing read per sequence on average at 0.50 bit nt$^{-1}$ (i.e., a sequencing depth of 1). Ostensibly, the low sequencing depth in this benchmark negated the benefits from clustering (as shown by the poor performance of the Yin-Yang codec) and thus enforced efficient use of the available read data. These results suggest read down-sampling is the only experimental benchmark sufficiently informative for codec comparisons out of the three literature experiments tested.

### Benchmarking codecs across common experimental workflow choices

From an experimental perspective, three parameters matter most for any DNA data storage workflow: the synthesis provider, the number of

**a** Pareto fronts of high- and low-fidelity scenarios

**b** Definition of high- and low-fidelity scenarios

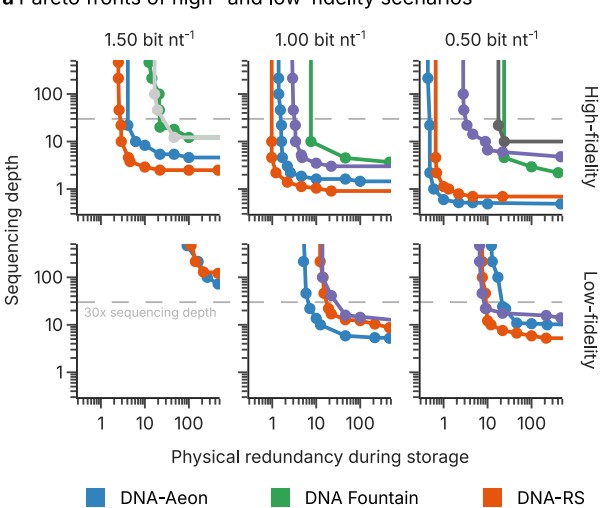

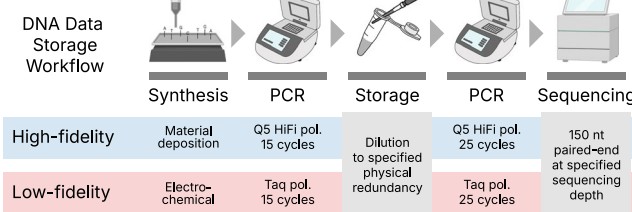

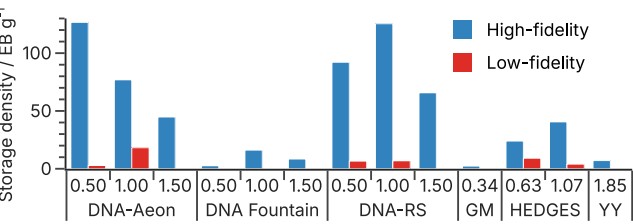

**c** Maximum storage density at 30x sequencing depth

**Fig. 3 | Codec performance in high- and low-fidelity scenarios. a** Pareto fronts for each codec and code rate in the high- (top) and low-fidelity scenario (bottom). The physical redundancy during storage (e.g., average number of oligos per sequence) and the sequencing depth (e.g., average number of reads per sequence) were varied to identify feasible and infeasible regions. Only Pareto efficient points are shown, and connected for illustration, with the feasible region lying above the curves. The dashed line highlights a sequencing depth of 30x, as used in the in vitro experiment. **b** Illustration of the high- and low-fidelity scenarios, resembling common workflows in DNA data storage. The physical redundancy and sequencing depth were systematically varied to obtain the Pareto fronts in (**a**). **c** Highest feasible storage density in the high- (blue) and low-fidelity (red) scenarios by codec and code rate at a sequencing depth of 30x (dashed lines in (**a**)). Storage densities only consider the payload, and assume a molecular weight of $662 \text{ g mol}^{-1} \text{ bp}^{-1}$, see Supplementary Note 2. **b** partially created in BioRender. Gimpel, A. (2025) https://BioRender.com/9wrsaez. Source data are provided as a Source Data file.

oligos per sequence present during storage ("physical redundancy"), and the number of sequencing reads per sequence ("sequencing depth")[13,16,17]. While the choice of synthesis provider is usually well-documented in literature studies and its effect on error rate well-established, the choices of physical redundancy and sequencing depth vary considerably across studies and affect only the homogeneity of sequencing reads[13,16,17]. Specifically, both low physical redundancies and low sequencing depths increase the likelihood of sequence dropout and diminish the diversity of the oligos due to the stochastic sampling of only few oligo copies. Importantly, physical redundancy directly affects the data's storage density (i.e., it is inversely proportional to storage density), whereas sequencing depth directly influences sequencing costs.

While the literature benchmarks presented above offer straightforward workflows, they still consider only the latter two parameters in isolation. Thus, we set out to compare the selected codecs across all three experimental parameters directly. First, we implemented two scenarios in DT4DDS (see Methods and Fig. 3b), centered around the two most-commonly used array-based synthesis technologies: material deposition/printing (i.e., Agilent, Twist Biosciences) and electrochemical synthesis (i.e., Genscript/CustomArray)[13]. The former is included in the high-uniformity, low-error scenario ("high-fidelity") using a high-fidelity polymerase for amplification and yielding an error rate of around 0.1% with minor sequence loss (<1% without dilution)[17]. In contrast, the low-uniformity, high-error scenario ("low-fidelity") includes electrochemical synthesis and amplification by an error-prone polymerase, yielding an error rate of around 1.5% and considerable sequence dropout (>2% without dilution)[17]. Within these two scenarios, we then varied the physical redundancy and sequencing depth akin to the synthetic scenario in Fig. 1e, thereby simultaneously optimizing for storage density and reading cost.

The Pareto fronts illustrating the tradeoff between physical redundancy and sequencing depth in the two scenarios are shown in Fig. 3a. Focusing first on the high-fidelity scenario (top half of Fig. 3a, average error rate 0.1%), a considerable separation of codecs was observed across all code rates. The Yin-Yang codec, as the baseline

without additional error-correction capabilities, showcases how clustering and the inherent repetition in sequencing reads suffice to yield a storage density of $6.6 \text{ EB g}^{-1}$ with a sequencing depth of 30× (32× physical redundancy at 1.85 bit nt$^{-1}$ code rate, see Fig. 3c). Both DNA Fountain ($15 \text{ EB g}^{-1}$ at 7.6× and 1.00 bit nt$^{-1}$) as well as HEDGES ($38 \text{ EB g}^{-1}$ at 3.2× and 1.07 bit nt$^{-1}$) considerably improved upon this baseline, despite their limited ability to tolerate considerable sequence dropout and large error rates simultaneously. In contrast, both DNA-Aeon and DNA-RS tolerated physical redundancies and sequencing depths below 10x even at a code rate of 1.50 bit nt$^{-1}$. At a code rate of 0.50 bit nt$^{-1}$, these codecs required only around one physical copy and one sequencing read per reference sequence on average, in-line with the results in Fig. 2. As a result, both DNA-Aeon and DNA-RS achieved storage densities well above the current state-of-the-art ($28 \text{ EB g}^{-1}$, see Supplementary Note 2 and Organick et al.[25]) in this scenario (about $117 \text{ EB g}^{-1}$ at 30× sequencing depth, see Fig. 3c).

Expectedly, the low-fidelity scenario (bottom half of Fig. 3a, average error rate 1.5%) and the significantly increased error load under these conditions, challenged codecs considerably more. Thus, required sequencing depths and physical redundancies were about 1–2 order of magnitude larger than in the high-fidelity scenario. Surprisingly, the DNA Fountain, Goldman, and Yin-Yang codecs were unable to decode the data in the low-fidelity scenario at all. As these codecs tolerated much larger average error rates in the synthetic benchmarks (see Fig. 1e, dotted line), this observation further questions the transferability of synthetic results to experimental workflows. The other codecs—DNA-Aeon, DNA-RS, and HEDGES—achieved generally similar performance, albeit with slight advantages for DNA-Aeon at 1.00 bit nt$^{-1}$ and DNA-RS at 0.50 bit nt$^{-1}$. Nonetheless, the overall performance falls drastically short compared to the storage densities obtained in the high-fidelity scenario, peaking at $17 \text{ EB g}^{-1}$ for DNA-Aeon at 1.00 bit nt$^{-1}$ (6.7× physical coverage, see Fig. 3c). Additionally, both DNA-Aeon and DNA-RS were partially limited by the time constraint, as shown in Supplementary Figs. 12, 13. This suggests that also in this benchmark, high error-correction capability is in a direct trade-

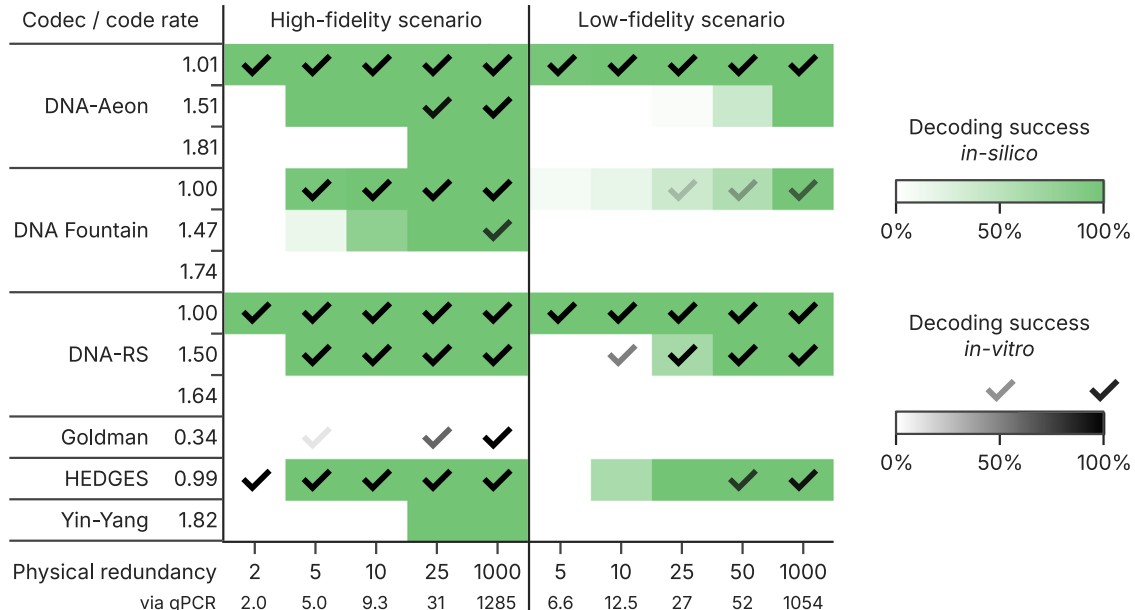

**Fig. 4 | Codec performance in the in vitro experiments.** The in vitro experiments followed the high- (left) and low-fidelity (right) workflows outlined in Fig. 3b with slight changes to the amount of PCR cycles (see Methods). Five nominal physical redundancies ranging from 2× to 1000× were implemented by dilution and validated by qPCR. For each codec and code rate, decoding success in both experimental workflows was assessed by each codec's ability to decode the experimental sequencing data after down-sampling to a sequencing depth of 30x (check marks). Robustness of decoding was assessed by performing a total of ten downsampling repetitions, with the fraction of successful repetitions reported (intensity of check marks). For comparison, the probabilities of decoding success estimated from replicating the amended high- and low-fidelity workflows in silico are also shown (see Methods, green shading). Source data are provided as a Source Data file.

off with computational power. Expectedly, memory use did not limit codec performance in either scenario.

Notably, most Pareto fronts in both scenarios feature a clear symmetry with respect to the extremes of physical redundancy and sequencing depth. This suggests down-sampling sequencing data to assess minimum sequencing depth—one of the literature benchmarks tested above – is a good proxy for the minimum physical redundancy required during storage. However, the Pareto fronts in Fig. 3a demonstrate clearly that the validity of this approach relies upon the other parameter being non-limiting. For example, while DNA-RS at 1.00 bit nt$^{-1}$ exhibited a minimum sequencing depth of 1× in the read down-sampling (see Fig. 2), recovering data with this codec after storage at a 1× physical redundancy would require a sequencing depth exceeding 20×.

Interestingly, the highest storage densities of DNA-Aeon (140 EB g$^{-1}$, 0.50 bit nt$^{-1}$, physical redundancy 0.49x) and DNA-RS (125 EB g$^{-1}$, 1.00 bit nt$^{-1}$, physical redundancy 0.97×) at 30× sequencing depth were not achieved at the highest code rate tested (i.e., 1.50 bit nt$^{-1}$). This suggests the major benefit of additional logical redundancy—enabling further reduction in physical redundancy – considerably outweighs the loss in code rate. However, choosing lower code rates increases synthesis costs proportionally, as the price of synthesis scales directly with the number of sequences to be synthesized resp. the amount of data stored (see Supplementary Fig. 20a, b). Given the standardized code rates used in this study, all codecs feature similar costs per bit stored, at 14 USD/kB on average in the high-fidelity scenario and 7.9 USD/kB in the low-fidelity scenario (at a 1 MB scale, see Supplementary Fig. 20c). Ultimately, synthesis cost is not directly defined by the choice of codec, but purely by the employed code rate. To this end, the choice of code rate must ultimately compromise between achieving the highest storage densities at lower code rates and reducing the synthesis costs at higher code rates. Thus, given the direct scaling of synthesis costs with code rate, our performance analysis comparing three standardized code rates also translates directly to anticipated synthesis costs.

## Experimental replications demonstrating state-of-the-art storage densities

Experimental replications of the high- and low-fidelity scenarios in vitro were performed to establish the accuracy of our in silico analysis and confirm the validity of our observations regarding storage density. For this, adjusted codec parameters were used to store files with 5 kB, 17 kB, and 19 kB into a total of 11 293 sequences using all six selected codecs. These parameters (detailed in Supplementary Tables 5–9) were chosen to target code rates of 1.00 bit nt$^{-1}$, 1.50 bit nt$^{-1}$, and the highest code rate supported by each codec, while adhering to a length limit of 126 nt (a constraint of electrochemical synthesis, see Methods). After synthesis by Genscript and Twist Biosciences, both oligo pools were amplified, diluted, reamplified, and sequenced as outlined in Fig. 3b for the synthetic scenarios. Figure 4 shows the decoding results of these in vitro replications of the low- and high-fidelity workflows, using five different physical redundancies at a fixed sequencing depth of 30x. The decoding results predicted from our in silico analysis are also shown for comparison (green shading).

Overall, the experimental results (see Fig. 4) closely follow the trends observed in our in silico analysis of the workflows (see Fig. 3a). Expectedly, the low-fidelity scenario proved more challenging than the high-fidelity scenarios across all codecs. Nonetheless, DNA-Aeon and DNA-RS performed best, both capable of recovering the data stored at physical redundancies as low as 2× in the high-, and 5× in the low-fidelity scenario (at a code rate of 1.0 bit nt$^{-1}$). Taking into account the physical redundancies measured by qPCR (see Fig. 4 and Supplementary Tables 13,14), this equals storage densities of 57 EB g$^{-1}$ and 17 EB g$^{-1}$, respectively (43 EB g$^{-1}$ and 13 EB g$^{-1}$ including adapters, see Supplementary Table 11). These results not only validate the accuracy of our benchmarks in Fig. 3, but also considerably improve upon previously demonstrated achievable data densities for DNA data storage (26 EB g$^{-1}$ using high-fidelity workflow by Organick et al.[25], 0.033 EB g$^{-1}$ using low-fidelity workflow by Grass et al.[7], see Supplementary Table 11).

High-density data storage in the high-fidelity scenario was also achieved with HEDGES (physical redundancy of 2×, equivalent to 56 EB g$^{-1}$). However, the HEDGES codec was less capable in the low-fidelity scenario, maxing out at a physical redundancy of 50× (52× by qPCR, equivalent to 2.2 EB g$^{-1}$). More generally, most codecs failed to decode the data in the low-fidelity scenario at code rates above 1.0 bit nt$^{-1}$, even at 1000× physical redundancy (see Fig.4, right). The DNA-RS codec is the only exception in our test, achieving successful decoding with 1.50 bit nt$^{-1}$ down to 10× physical redundancy (12.5× by qPCR, 14 EB g$^{-1}$). This highlights the ability of the low-fidelity scenario to differentiate codecs by testing their tolerance to both errors and sequence dropout. In contrast, all codecs with error-correction capabilities worked reliably at 1.5 bit nt$^{-1}$ and 1000× physical redundancy in the high-fidelity scenario.

Figure 4 also shows the probability of successful decoding derived for the experimental conditions from in silico simulations. These decoding probabilities show good agreement with the experimental results, highlighting the overall accuracy of the in silico simulations. However, systematic deviations to experimental results were present in some cases. In the high-fidelity scenario (Fig. 4, left), the capabilities of the DNA-Aeon, DNA Fountain and Yin-Yang codecs were overestimated. In contrast, these simulations were too pessimistic with respect to the HEDGES and Goldman codecs. The predictions in the low-fidelity scenario (Fig. 4, right)—while generally more accurate—tended to slightly overestimate the capabilities of the DNA-Aeon and HEDGES codecs.

To elucidate the source of these systematic deviations, the homogeneity of error rates and sequence dropout in the experimental datasets was assessed. While the rates of errors were similar across codecs, the rate of sequence dropout varied drastically between codecs at the same physical redundancy (see Supplementary Fig. 15). Upon closer inspection, these variations were caused by systematic differences in the coverage homogeneity of each codec's sequences (see Supplementary Figs. 16, 17). Evidently, the sequences generated by some codecs were less homogeneously represented in the oligo pools than others, with coefficients of variation at 1000× physical coverage ranging from 0.38 to 1.15. As all sequences were synthesized on the same chips in a randomized order, this inhomogeneity does not impact the fairness of the experiments. Instead, the differences in pool homogeneity likely resulted from biases during amplification, potentially caused by certain sequence features[37,38]. For example, the sequences generated by the Goldman codec were least abundant in both scenarios, at half the mean sequencing depth (see Supplementary Fig. 15). Here, the repetitive elements introduced by the codec could inhibit amplification by enabling secondary structures[39]. In line with this reasoning, its sequences are underrepresented in the sequencing data (see Supplementary Fig. 15), and thus less likely to be sampled during dilution. This explains the increased rates of sequence dropout from these codecs, which cause the systematic deviations to the simulation results (additional discussion is provided in Supplementary Note 3). However, these deviations in copy numbers in the physical pool do not explain the much larger differences between codec performances across physical redundancies in Fig. 4.

### Extension of benchmarking pipeline to additional literature codecs

Finally, to highlight the utility of the developed benchmarking pipeline for codec comparisons, the scope of literature codecs was extended to cover additional codecs by Song et al. (DBGPS)[19], Chandak et al. (LDPC)[40] and Zan et al. (Modulation)[41]. Figure 5 shows the results of the in silico benchmarks for these three additional codecs, with the results by the best-performing codecs from the main study—DNA-Aeon by Welzel et al.[11] and DNA-RS by Heckel et al.[18,28] —also shown for comparison. In line with the results from Fig. 1d, clustering by CD-HIT

drastically improved error tolerance across these additional codecs, see Fig. 5a. Moreover, these codecs' ability to correct individual error types did not differ from the results of DNA-Aeon and DNA-RS (see Fig. 5b).

Importantly, the Pareto fronts across code rates in Fig. 5c, d show that DBGPS by Song et al.[19] performed similarly to DNA-Aeon and DNA-RS in most cases. Thus, its performance is on-par with these state-of-the-art codecs, to within the experimental error associated with the lack of parameter optimization in this study. In contrast, both LDPC by Chandak et al.[40] and the modulation-based codec by Zan et al.[41] fall behind the performance of DNA-Aeon, DNA-RS, and DBGPS in our silico benchmarks.

In the case of the modulation-based codec by Zan et al.[41], its lack of data-level error correction renders it comparable to the Yin-Yang codec by Ping et al.[24] investigated in our initial codec benchmark. Reassuringly, both of these codecs relying solely on clustering and sequence alignment for error correction achieved similar tolerated error rates of around 4–5% in our test (compare Figs. 1d, 5a). However, contrary to the Yin-Yang codec by Ping et al.[24], the implementation of the modulation-based codec provided in the repository by Zan et al.[41] does not include sequence indexing. As a result, it cannot recover the stored data if sequences are recovered out of order, unless sequence indexing is implemented separately from the codec. Naturally, features such as sequence indexing and an outer code for tolerance to sequence loss can be retrofitted to existing codecs, using established methods like Fountain or Reed-Solomon codes. However, such adaptions to existing codec implementations require additional programming, testing and optimization efforts, which is not realistic for most use cases and was therefore not in scope of this study. These limitations of the modulation-based codec cause the discrepancy in observed performance between Zan et al.[41] and the benchmark in Fig. 5.

Taken together, the extension of our in silico pipeline to three additional codecs from the literature demonstrates its simplicity and utility for codec benchmarking, without requiring any experimental effort. In addition, the results obtained for the DBGPS codec by Song et al.[19] further support the observation that a well-executed, balanced inner/outer code separation strategy is more important for codec performance than the specific choice of employed error-correction code. Thus, these results showcase both how error-correction codecs for DNA data storage are optimally constructed and how their performance is realistically assessed.

## Discussion

The comparison of error-correction coding for DNA data storage in this work highlights the maturity of the field while closing in on DNA's physical limits. To do so, this study systematically harmonized six established codecs across three code rates to perform fair performance comparisons in standardized scenarios. This study thereby comprehensively established and experimentally verified the current state-of-the-art in error-correction coding for DNA data storage under unbiased and standardized conditions.

Using synthetic error benchmarks, this study first isolates codec performance from experimental factors by assessing codecs' tolerance to errors and sequence dropout both individually and simultaneously. Notably, we find that existing codecs can tolerate error rates up to 14% and sequence loss as high as 65% in isolation, but codecs' abilities to correct both simultaneously varies drastically. Then, using literature experiments replicated in silico, we identify shortcomings in commonly performed experimental benchmarks. Taken together, our results thus challenge common practice in the DNA data storage literature, as both synthetic error benchmarks and serial dilution/amplification experiments are widely-used standards to showcase codec performance[10,11,23,24], but fail to differentiate codec performance in our standardized setup.

**a** Effect of codec–clustering pairing

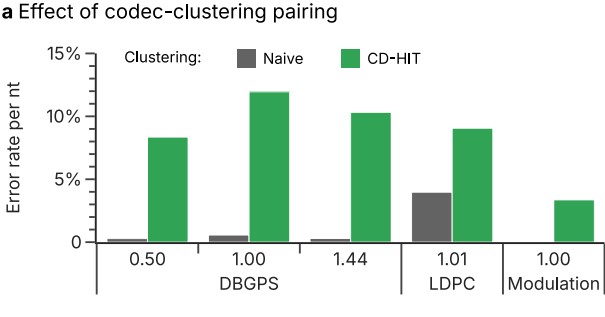

**b** Error tolerances against individual error types

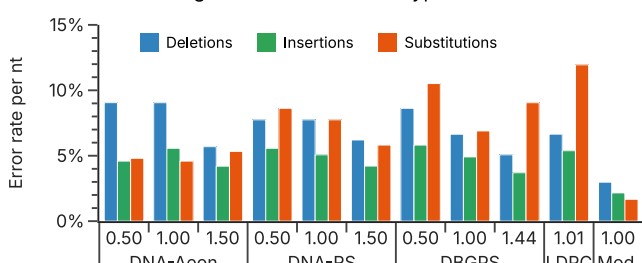

**c** Pareto fronts between errors and sequence dropout

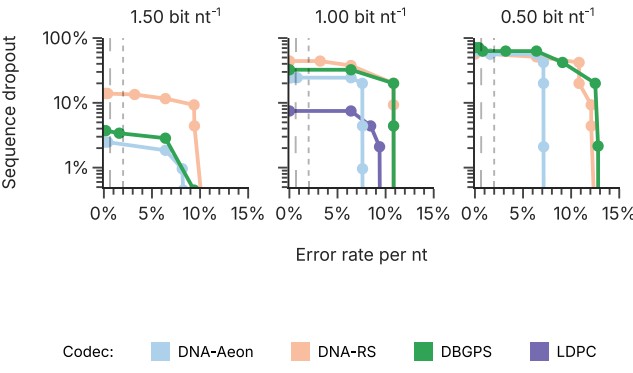

**d** Pareto fronts of high- and low-fidelity scenarios

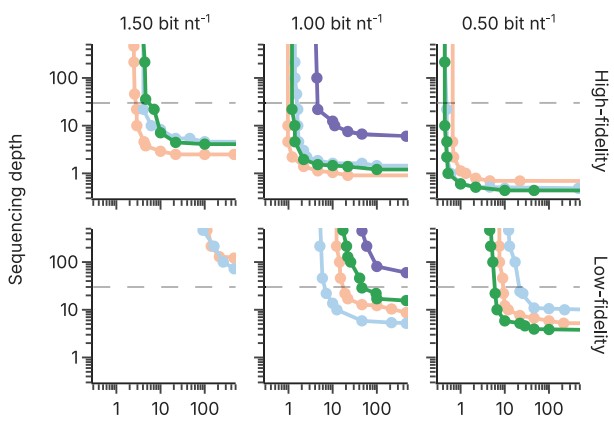

**Fig. 5 | Extension of the in silico analysis to additional literature codecs. a** The effect of clustering on the additional literature codecs by Song et al. (DBGPS)[19], Chandak et al. (LDPC)[40] and Zan et al. (Modulation)[41], as measured in tolerated error rate, between naïve clustering (gray) and the best-performing clustering algorithm (green, CD-HIT). The error rate shown corresponds to the combined rate of substitutions, deletions, and insertions at a fixed composition of 53% substitutions, 45% deletions, and 2% insertions, thereby resembling the experimental error pattern in ref. 17. **b** Comparison of each codec's error tolerance against individual error types (deletions, blue; insertions, green; substitutions, orange). **c** Pareto fronts of the additional codecs upon simultaneous consideration of errors and sequence dropout, grouped by code rate. **d** Pareto fronts of the additional codecs in the high- (top row) and low-fidelity scenarios (bottom row) upon variation of the initial coverage and the sequencing depth. In the high-fidelity scenario, synthesis by material deposition and amplification with a high-fidelity polymerase is assumed, whereas the low-fidelity scenario includes electrochemical synthesis and amplification with Taq polymerase. In panels c) and d), the best-performing codecs from the main study—DNA-Aeon by Welzel et al.[11] and DNA-RS by Heckel et al.[18,28]—are shown in lighter colors for comparison. Source data are provided as a Source Data file.

As an alternative, rigorous benchmark, we propose and implement both a high- and low-fidelity scenario, thereby systematically assessing codecs across three critical experimental factors: synthesis technology, physical redundancy, and sequencing depth. Importantly, we show that only little logical redundancy is required for high-density storage (i.e., up to 7 EB g$^{-1}$) using the common high-fidelity scenario (i.e., synthesis by Twist Biosciences and amplification with high-fidelity polymerases). In contrast, enabling data storage at low physical redundancy and low sequencing depth required a balanced inner/outer code separation strategy (i.e., DNA-Aeon, DNA-RS, and DBGPS), theoretically enabling storage densities as high as 117 EB g$^{-1}$ (see Fig. 3c). Extension of our synthetic benchmark suite to three additional literature codecs further highlighted that state-of-the-art performance depended more on a well-executed, balanced inner/outer code separation strategy than on the specific choice of error-correction code. This suggests existing codecs are sufficiently close to the theoretical limit, such that minor implementation details (e.g., optimality of parameter choices, balance of inner and outer redundancy) cause the remaining, minor gaps in codec performance in most cases.

An experimental replication of our rigorous benchmark constitutes an objective, experimental benchmark of standardized codecs from the literature, using multiple code rates and both a high- and low-fidelity scenario. We experimentally demonstrate data storage at 43 EB g$^{-1}$ and 13 EB g$^{-1}$ in the high- and low-fidelity scenarios respectively, using only these established codecs from the literature. These results mark a major step towards the theoretical limit for data storage in DNA, at 227 EB g$^{-1}$ (see Supplementary Note 2), first postulated by Church et al.[5] in 2012. Moreover, they improve upon both the first experimental investigation of storage density by Erlich and Zielinski[10] in 2017, at 215 PB g$^{-1}$, and the subsequent improvement by Organick et al.[25] to 17 EB g$^{-1}$ in 2020. Additionally, our experimental demonstration of data storage at 13 EB g$^{-1}$ in the low-fidelity scenario represents the first experimental investigation of maximum storage density using error-prone electrochemical DNA synthesis. In doing so, our approach to experimental benchmarking ensured an unbiased comparison and precluded any discrimination against a codec by the experimenter.

All in all, this work provides strong evidence against the use of non-standardized, synthetic error benchmarks as the foremost performance indicators for codecs. Instead, codec comparisons in the future should harmonize codec parameters, test for tolerance against errors and sequence dropout simultaneously, and include a comparison with the state-of-the-art. Based on our results, this would include at least a standardized read down-sampling experiment (either in silico, e.g., with DT4DDS[17], or ideally in vitro, if possible), comparing against either DNA-Aeon[11], DNA-RS[18,28,29], or DBGPS[19] as the state-of-the-art.

Beyond raw codec performance, decoding speed also emerged as another major limitation throughout this study. The analysis of the

computational performance throughout all benchmarks in this study (see Supplementary Figs. 4–7 and 9–13) shows that only DNA-Aeon, DNA-RS, and HEDGES were limited by the runtime constraint at all. However, runtime generally increased sharply as error rates approached each codec's limits, such that, in many cases, codecs failed to successfully decode the data before being limited by our time constraint. Nonetheless, codecs' runtime and parallelizability should become an important metric for consideration as data volumes for DNA data storage continue to increase.

Our analysis shows that codec comparison and benchmarking is important for discussing new coding schemes, and our in silico pipeline contributes suitable benchmarking tools to the current literature. In addition, our analysis demonstrates that although DNA data storage represents an error channel with error patterns different to preceding data storage technologies, error-correcting principles developed for existing channels remain highly relevant. Consequently, the state-of-the-art codes for DNA data storage leveraging existing Reed-Solomon-, Fountain- and LDPC-codes perform best if in-sequence error correction is combined with approaches to reconstruct lost sequences and compensate for remaining errors within the sequences. An additional advantage of using pre-existing coding principles is the certainty that such codes can be brought to production level coding/decoding speeds with appropriate implementation into dedicated hardware, as e.g. in CD-ROM drives[42]. Under the DNA data storage scenarios investigated in this work, constrained coding approaches do not have competitive edge over non-constrained coding, as also discussed in detail by Weindel et al.[43]. However, better codes for correcting deletions and insertions might be beneficial for future DNA storage systems and allow data reconstruction in error prone scenarios even without the current aligning of multiple reads for insertion/deletion correction.

A major limitation of this work is the small selection of investigated codecs and clustering algorithms, as well as the lack of a systematic optimization of their parameters. For example, recent efforts to employ deep learning in DNA data storage[26,44,45] were not covered in this study, but could benefit from the benchmarking scheme proposed in this study to comprehensively assess their scalability and transferability. Moreover, the investigated scenarios omit less common workflows, such as photolithographic[18,46] or enzymatic[47,48] synthesis, nanopore sequencing[21,22], aging-induced decay[19,49], or the use of degenerate bases[23,50,51]. In these workflows, certain codec features (e.g., constraints, error type specificity) could convey advantages which were not evident in the common workflows used in this study. For example, only the DNA-Aeon codec can be constrained to avoid biological motifs (see Table 1), which can negatively affect codec performance if motif-containing sequences are considerably more prone to errors or dropout (see Supplementary Fig. 19).

In addition, as DNA data storage might gravitate towards even lower-fidelity processes to decrease costs, new challenges for error-correction coding might open up in the future. This shift from historically high-fidelity processes—requiring little to no logical redundancy in the first proof-of-concept studies by Goldman et al.[6] and Church et al.[5] — towards cost-effective, low-fidelity processes will require the use of reliable, highly error-tolerant codecs. To this end, this study suggests codecs using a balanced inner/outer code separation strategy, such as DNA-Aeon, DNA-RS, or DBGPS, are best-suited to fill this gap. In all cases, the standardized approach to benchmarking presented in this work—ensuring fairness by eliminating biases from experimental errors and experimentalists' preferences—establishes suitable best practices for future codec comparisons in DNA data storage.

## Methods
### Selection of codecs and clustering algorithms
The selection of codecs for this study was based on the availability of an open-source implementation with sufficient documentation, the presence of in vitro experiments in the original publication, the prominence in literature, and considerations for covering a broad range of approaches. We therefore selected DNA-Aeon by Welzel et al.[11], DNA Fountain by Erlich and Zielinski[10,27], DNA-RS by Heckel[18,28,29], an implementation of the codec used by Goldman et al.[6] ("Goldman"), HEDGES by Press et al.[30], and Yin-Yang by Ping et al.[24] (without additional error-correction elements). The installation and usage for encoding and decoding of each codec followed the documentation as far as possible. Nonetheless, some minor changes to the implementations of the DNA-Aeon, Goldman, HEDGES, and Yin-Yang codecs were required to either facilitate automated testing or ensure representative performance. These changes are described in Supplementary Note 1 and are available in the code associated with this study.

The selection of clustering algorithms for this study was based on the same criteria as for the codecs. However, due to the sparse availability of suitable clustering algorithms specific to DNA data storage, we mainly considered general-purpose clustering algorithms. We therefore selected CD-HIT by Li et al.[31,52,53], Clover by Qu et al.[34], clustering based on Locality-Sensititve Hashing by Darestani and Heckel[18,54] ("LSH"), MMseqs2 by Steinegger et al.[32,55], and Starcode by Zorita et al.[33,56]. In addition, we implement the naïve clustering approach used for DNA Fountain[10] which simply uses the unique sequencing reads sorted by their abundance.

**Selection of codec parameters.** All of the selected codecs which support adjusting the sequence design and/or the level of error-correction were harmonized with respect to code rate, sequence length, and constraint choice. If supported, three sets of parameters yielding code rates of 1.50 bit nt$^{-1}$, 1.00 bit nt$^{-1}$, and 0.50 bit nt$^{-1}$ at a sequence length of around 150 nt were created for the in silico studies. For the in vitro experiment, due to the constraint on sequence length imposed by one of the synthesis providers (Genscript, 170 nt including adapters), three sets of parameters yielding code rates of 1.50 bit nt$^{-1}$, 1.00 bit nt$^{-1}$, and the highest code rate possible, using at most 126 nt, were selected. A detailed list of codec parameters used in this study is provided in Supplementary Tables 5–9.

### Benchmarking of clustering algorithms
To compare the selected clustering algorithms individually and select their optimal parameters, two experimental sequencing datasets from a previous study[17] (PRJEB65931) were used. To preclude any impact of sequence design on clustering performance, these sequencing datasets were obtained from randomly generated sequences. To test both a high- and a low-fidelity scenario, both an experiment with synthesis by material deposition (ERR12033821) and one by electrochemical synthesis (ERR12033820) were used. For more details, see ref. 17. Prior to clustering, the paired sequencing reads were merged with NGmerge[57] (v0.3) and subsampled to a sequencing coverage of 20 sequencing reads per reference sequence using seqtk[58] (v1.4). After clustering, individual clusters were aligned using Kalign[59] (v3.4.0) to yield consensus sequences.

Besides clustering speed, three other performance metrics were calculated by comparing the consensus sequences after clustering to the reference sequences. To do so, each consensus sequence was assigned to a reference sequence by minimizing their Levenshtein distance. First, sensitivity was defined as the fraction of reference sequences which still had at least one associated consensus sequence after clustering. Second, accuracy was defined as the mean Levenshtein similarity of the closest match to each reference sequence. Third, specificity was defined as the ratio of the number of reference sequences with at least one associated consensus sequence relative to the total number of consensus sequences.

## Setup of the simulation pipelines for codec evaluation

Each simulation pipeline consisted of the steps outlined in Fig. 1a: encoding, workflow, clustering, and decoding. For encoding, a set of fixed parameters (see below) for each codec was used to encode a 19 kB binary file with random content into DNA sequences ("reference sequences"). The workflow, either a script to introduce random errors at fixed rates or a workflow implemented in DT4DDS[17,60] (v1.1, see below), then generated simulated sequencing reads from the reference sequences. These sequencing reads were then clustered by a specified clustering algorithm (see above), and the clusters were aligned individually using Kalign[59] (v3.4.0) to yield consensus sequences. Finally, the consensus sequences were provided to the codec together with any supplementary data if needed, in order to attempt decoding. Decoding success was assessed by byte-by-byte comparison to the original input file, with only complete recovery of the data being considered as successful decoding.

In all cases, the consensus sequences generated after clustering were padded or trimmed to the length of the reference sequences. This was necessary as several codec implementations were incompatible with sequences shorter or longer than those originally designed. Moreover, workflows using DT4DDS[17] (see below) —thus yielding paired sequencing reads—employed NGmerge[57] (v0.3) for read merging prior to clustering.

To automate the process of preparing and running the simulation pipelines, management tools and wrapper scripts were written in Python (v3.11) using BioPython (v1.84), scipy (v1.14.1), statsmodels (v0.14.1), numpy (v2.0.2), pandas (v2.2.3), plotly (v5.24.0), psutil (v6.0.0), RapidFuzz (v3.10.0), bamboost (v0.8.0), and h5py (v3.12.1) under Ubuntu 22.04 LTS.

**Computational constraints.** All simulation pipelines were run on the Euler cluster operated by the High-Performance Computing group at ETH Zürich, using the Slurm workload manager. Each pipeline was constrained to one core of an AMD EPYC 7763 CPU (2.45 GHz nominal, 3.50 GHz peak), 8 GB RAM (DDR4, 3200 MHz), and 2 GB of temporary disk space. Each individual step of a pipeline was further limited to one hour of runtime programmatically. If any constraint was violated throughout the pipeline, decoding was considered unsuccessful.

**Definition and estimation of decoding probability.** In order to assess the resilience of a chosen selection of codec, parameters, and clustering algorithm towards any of the workflow's experimental parameters, a one-dimensional sensitivity analysis was performed (see below). Using the binary outcomes of this analysis (i.e., decoding success at each parameter choice) as dependent variables, a logit model was fitted to estimate the parameter value at which the probability of successful decoding would equal 95%. This value was then considered the performance threshold for the chosen selection of codec, parameters, and clustering algorithm.

**One-dimensional sensitivity analysis.** Each one-dimensional sensitivity analysis was performed for a single parameter of a workflow and over a specified parameter range. To increase the accuracy of the estimated performance threshold, the sensitivity analysis was performed in three stages. In the first stage, ten logarithmically-spaced points were chosen across the full parameter range and tested. In the second stage, the ten outcomes of the first stage were used for a rough estimation of the performance threshold $t$ (see above), and ten additional logarithmically-spaced points were selected from the range $[\frac{t}{2}, 2t]$ and tested. In the third stage, the procedure of the second stage was repeated using the points from both the first and the second stage. The complete set of thirty binary outcomes were then used to generate the final estimate of the performance threshold (see above).

**Generation of Pareto fronts.** In order to enable sensitivity analyses across two parameters of a workflow, two sets of one-dimensional sensitivity analyses were performed. In each set, one of the workflow parameters was fixed to one of ten logarithmically-spaced values selected from the specified parameter range, while the other was varied according to the three-stage process outlined above. From the resulting combinations of feasible parameter thresholds, only the Pareto efficient points were used to generate Pareto fronts.

## In silico experiments

**Pairing of codecs with clustering algorithms.** For the pairing with codecs, only the best-performing parameter sets of each clustering algorithm from the individual benchmarking (see above) were considered. Each clustering algorithm was then tested with each codec in the basic error scenario, as described in detail in the following section. The clustering algorithm with the best performance threshold for each codec was then used throughout the rest of the study. The pairings are provided in Supplementary Table 3.

**Evaluation of codec performance in basic error scenarios.** A workflow was implemented which introduced specific error types at specified rates randomly throughout the reference sequences, generating 30 erroneous reads per sequence. Considered error types were substitutions (with equal probability for all substituting nucleobases), insertions (with equal probability for all inserted nucleobases), deletions, mixed-errors (at a fixed ratio of 53:45:2 substitutions:deletions:insertions, resembling the error pattern in ref. 17), and sequence dropout (i.e., the removal of a fixed proportion of reference sequences from the reads). All error types were considered in a range of $0.1$–$40\%$ $nt^{-1}$, and sequence dropout used a range of $0.5$–$99\%$. This workflow was used for the one-dimensional sensitivity analyses and the Pareto fronts in Fig. 1.

**Evaluation of codec performance in literature experiments.** The serial dilution and serial PCR experiments used in the study by Erlich and Zielinski[10], as well as the read down-sampling experiment by Organick et al.[9], were replicated as workflows using DT4DDS[17]. All workflows assumed high-fidelity synthesis by material deposition, and the use of a high-fidelity polymerase. Sequencing depths were adapted to reflect the average sequencing depths reported in the original studies. Contrary to the other workflows, the comparison of codecs in these replicated literature experiments did not use the sensitivity analysis approach described above. Instead, each iteration of the workflow was tested ten times and the highest iteration count with which all ten tests were successfully decoded was reported.

**Evaluation of codec performance in high- and low-fidelity scenarios.** Both a high- and a low-fidelity version of a generic data storage workflow, following the definitions for the best- and worst-case in Gimpel et al.[17], were implemented using DT4DDS[17]. These scenarios are outlined in Fig. 3b. In short, the simulated workflows include synthesis by material deposition or electrochemical synthesis, amplification for 15 cycles with either a high- (Q5) or low-fidelity (Taq) polymerase, dilution to a specified mean physical coverage, re-amplification for 25 cycles, and sequencing at a specified sequencing depth using paired-end, 150 nt reads of an iSeq 100. For both the high- and low-fidelity scenario, the mean physical redundancy after dilution and the sequencing depth were varied (see above) to yield the Pareto fronts in Fig. 3a.

**Replication of the in vitro workflows.** To compare the experimental data (see below) with the simulated results from the in silico analysis, the high- and low-fidelity scenarios used previously (see above) were adapted to use the adjusted codec parameters (see above) and match the workflow employed in the in vitro experiment. Specifically, the

deletion rate during synthesis in the low-fidelity scenario was decreased to 0.0044 nt$^{-1}$, the insertion rate during synthesis increased to 0.0010 nt$^{-1}$, and the number of PCR cycles increased to 23 and 29 in the first and second round of amplification, respectively. These adapted workflows were used as described above in a one-dimensional sensitivity analysis, varying the physical redundancy after dilution to yield the decoding probability as a function of physical redundancy.

### Experimental replication of high- and low-fidelity scenarios

The high- and low-fidelity scenarios outlined above were recreated experimentally with identically composed oligo pools synthesized by Twist Biosciences (South San Francisco, CA, United States) and Genscript (Piscataway, NJ, United States), using established protocols for DNA data storage[28]. As Genscript only supported a total sequence length of 170 nt including PCR adapters, the parameters of all codecs were adjusted (see above) to limit the reference sequences to 126 nt, leaving 41 nt for truncated Illumina TruSeq adapters and 3 nt for a codec-specific suffix (see Supplementary Fig. 14). In addition, to limit the number of sequences to be synthesized, the parameter sets yielding code rates of 0.50 bit nt$^{-1}$ were replaced with parameter sets yielding the maximum code rate supported by each codec. Multiple versions of a compressed image of ETH Zürich's main building (created by ETH Zürich/Gian Marco Castelberg) were used as input files, with either 5 kB (Goldman codec), 17 kB (all codecs at 1.00 bit nt$^{-1}$), or 19 kB (all others). As the sequence length varied between codecs, shorter reference sequences were padded with random nucleotides (see Supplementary Table 10 and Supplementary Fig. 14). In total, 11 293 sequences across all codecs were created, padded and indexed (up to 129 nt), supplied with PCR adapters (for a total of 170 nt), and their order randomized prior to being ordered for synthesis.

**Pool preparation.** The oligo pools ordered from Twist Biosciences and Genscript were handled and amplified individually. The oligo pool by Twist Biosciences, received dry, was resuspended to 10 ng μL$^{-1}$ with ultrapure water. To create a master pool, 1 μL of a 5000x dilution of the oligo pool was then amplified with 10 μL Q5 High-Fidelity polymerase master mix (New England Biolabs, M0492S), 1 μL of 10 μM 0 F and 0 R primers each (Microsynth, Balgach, Switzerland, see Supplementary Table 12), and 7 μL ultrapure water, replicated in a total of 96 wells. Thermocycling followed established protocols[28], using an initial denaturation at 95 °C for 3 min, followed by 15 cycles at 95 °C for 15 s, 54 °C for 30 s, and 72 °C for 30 s.

The oligo pool by Genscript, received as a solution, was diluted to 5 ng μL$^{-1}$ with ultrapure water. To create a master pool, 1 μL of the diluted oligo pool was then amplified with 10 μL KAPA SYBR FAST polymerase master mix (Sigma-Aldrich), 1 μL of 10 μM 0 F and 0 R primers each (Microsynth, see Supplementary Table 12), and 7 μL ultrapure water, replicated in a total of 96 wells. Thermocycling followed the aforementioned protocol for 23 cycles.

For both oligo pools, each pool's wells were then combined and purified (DNA Clean & Concentrator-5, ZymoResearch). To increase purity further, each pool was then run on an agarose gel (E-Gel EX Agarose Gels 2%, Invitrogen) and the appropriate bands excised and purified (ZymoClean Gel DNA Recovery Kit, ZymoResearch). Finally, the pools were dialyzed (0.025 μm, 25 mm VSWP membrane, MF-Millipore) for 4 hours against ultrapure water. Concentration was measured by fluorescence (Qubit dsDNA HS Kit, Invitrogen) and by spectrophotometry (NanoDrop, Thermo Scientific).

The concentration of the master pool prepared from Twist Biosciences was measured as 49.4 ng μL$^{-1}$ (Qubit) and 51.5 ng μL$^{-1}$ (NanoDrop) respectively. Its concentration was therefore averaged to 50.45 ng μL$^{-1}$ for all further experiments. The concentration of the master pool prepared from Genscript was measured as 20.6 ng μL$^{-1}$

(Qubit) and 18.2 ng μL$^{-1}$ (NanoDrop) respectively. Its concentration was therefore averaged to 19.40 ng μL$^{-1}$ for all further experiments.

**Dilution and quantification by qPCR.** Dilution to specified physical coverages (1000x, 50x, 25x, 10x, 5x, and 2x) was performed starting from the master pool. To convert from specified physical redundancy to required concentration, 11,293 sequences of dsDNA with 170 nt were assumed, yielding a physical redundancy of 509074× per ng. All dilutions were prepared such that 5 μL contained the required mass corresponding to the specified physical redundancy $r$, e.g., the concentration $c$ was selected as $c = \frac{r}{5\mu L \cdot 509074\, ng^{-1}}$ Dilutions were performed serially while limiting the dilution factor to below 100x in each step, thereby maximizing dilution accuracy.

Calibration curves for qPCR were set up for each master pool individually, using serial dilutions of the master pools spanning a range from 0.05 ng μL$^{-1}$ (equivalent to a coverage of around 130,000 per 5 μL) to 5·10$^{-8}$ ng μL$^{-1}$ (coverage of 0.13 per 5 μL). qPCR used the same thermocycling settings as above, using 5 μL sample with 10 μL KAPA SYBR FAST polymerase master mix (Sigma-Aldrich), 1 μL of 10 μM 0 F and 0 R primers each (Microsynth, see Supplementary Table 12), and 3 μL ultrapure water, measured in duplicates. The calibration curves are shown in Supplementary Fig. 18. The qPCR results of all prepared dilutions, calibrated against a standard from the calibration curve measured in parallel, are given in Supplementary Tables 13 and 14.

**Sequencing.** Sequencing preparation followed established protocols[17,28]. In short, 5 μL of each dilution was amplified for 20 cycles (samples of Genscript and Twist pool with 1000× physical redundancy), 26 cycles (only Genscript pool, 50× and 25× physical redundancy), or 29 cycles (all others) with 10 μL KAPA SYBR FAST polymerase master mix (only Genscript pool, Sigma-Aldrich) or 10 μL Q5 High-Fidelity polymerase master mix (only Twist pool, New England Biolabs, M0492S), 1 μL of 10 μM 2FUF and indexed 2RIF primers each (Microsynth, see Supplementary Table 12), and 3 μL ultrapure water. Amplified samples were run on an agarose gel (E-Gel EX Agarose Gels 2%, Invitrogen) and the appropriate band excised and purified (ZymoClean Gel DNA Recovery Kit, ZymoResearch) prior to quantification by fluorescence (Qubit dsDNA HS Kit, Invitrogen). All purified samples were individually diluted to 1 nM and pooled. The combined samples, diluted to 50 pM, were added to an iSeq 100 i1 Reagent v2 cartridge (Illumina) for 150 nt paired-end sequencing.

**Analysis of error rates and coverage biases.** Sequence coverage in all sequencing datasets was assessed by read mapping with BBMap[61] (v39.01). Error analysis of the sample with 1000× coverage was performed as outlined in Gimpel et al.[17] using the tools implemented in DT4DDS[17]. Error analysis of the sets of sequences belonging to a single codec at a specified code rate were performed by using only the reference sequences of that codec for mapping with BBMap.

**Evaluation of codec performance in the in vitro experiment.** The sequencing reads corresponding to each sample with a specified physical coverage were separated into subsets for each codec and code rate by filtering with BBMap[61]. After separation, each set of sequencing reads was randomly downsampled to a sequencing depth of 30 reads per reference sequence ten times. These sampled sequencing reads were used as input to the decoding pipeline consisting of a codec-specific clustering algorithm (see above) and the decoding step of the codec itself. Decoding success was assessed by byte-by-byte comparison to the original input file used for encoding.

### Reporting summary

Further information on research design is available in the Nature Portfolio Reporting Summary linked to this article.

## Data availability

The sequencing data generated in this study has been deposited in the European Nucleotide Archive under accession code PRJEB90546. Source data are provided with this paper.

## Code availability

The code for benchmarking simulations and the Jupyter Notebooks for data analysis are publicly available and have been deposited in the public GitHub repositories at github.com/fml-ethz/dt4dds-benchmark[62] [https://doi.org/10.5281/zenodo.17391326] and github.com/fml-ethz/dt4dds-benchmark_notebooks[63] [https://doi.org/10.5281/zenodo.17391328], under a GPLv3 license. The specific versions of the code associated with this publication are archived in Zenodo and accessible via https://doi.org/10.5281/zenodo.17391326 and https://doi.org/10.5281/zenodo.17391328.

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

## Acknowledgements

This project was financed by the European Union's Horizon 2020 Program, FET-Open: DNA-FAIRYLIGHTS, grant agreement no. 964995, and the European Union's Horizon EIC Pathfinder Challenge Program: DiDAX, Grant Agreement No. 101115134 (Swiss Participants supported by the Swiss Secretariat for Education, Research and Innovation (SERI) under contract number 23.00330). Views and opinions expressed are, however, those of the authors only and do not necessarily reflect those of the European Union or the European Research Council Executive Agency. Neither the European Union nor the granting authority can be held responsible for them. Data analysis and simulations were performed on the Euler cluster operated by the High-Performance Computing group at ETH Zürich. Figures were partially created with BioRender.com (see BioRender. Gimpel, A. (2025) https://BioRender.com/9wrsaez) and icons for Figs. 1, 2, and 3 (conical microtube, pipette, and thermocycler) were provided by Labicons (www.labicons.net).

## Author contributions

R.N.G. and A.L.G. initiated and supervised the project with input from W.J.S. and R.H. A.R. and A.L.G. performed the experiments. A.R. and A.L.G. developed the code, and performed simulations as well as data analysis. A.L.G. prepared the illustrations, and wrote the manuscript with input and approval from all authors.

## Funding

## Competing interests

W.J.S., R.H., and R.N.G. are authors of the studies demonstrating the use of the DNA-RS codec and the LSH clustering algorithm[18,28]. The other authors declare no competing interests.
