## [Transparent Peer Review file · Nature Communications]

Comparison of state-of-the-art error-correction coding for sequence-based DNA data storage

Corresponding Author: Professor Robert Grass

Version 0:

Reviewer comments:

Reviewer #1

(Remarks to the Author)

DNA storage is emerging as a potential alternative to the challenge of exponentially increasing data. One major topic is how to effectively retrieve information from the IDS error caused by DNA synthesis, PCR and sequencing process. and the dropout issue in the random sampling process is also a critical issue to be solved. Up to now, many methods have proposed to tackle these problems. this paper intended to make a comprehensive comparison of some famous codec to evaluate their performance. Although this work is very meaningful for us to understand the real performance of these methods, there exist some issues need to be solved:

1. the dropout issue is a unique problem in DNA storage in the random sampling process, it is relative easy to solve by the fountain code which is very mature in tradition communication society. And any codec could use it to solve this issue. therefore, the key issue should be the IDS error correction.

my suggestion is to separate the evaluation of IDS error correction capability and dropout issue.

2. what is the criterion of the six SOTA methods? I suggest the author should select more representative methods:

(1) The modulation-based codec which can tolerate high IDS errors(<https://pubs.acs.org/doi/abs/10.1021/acs.jcim.3c00629>)

(2) deep learning based methods(<https://onlinelibrary.wiley.com/doi/abs/10.1002/smt.202400959>,
<https://www.nature.com/articles/s41598-024-83806-5>)

3. the Yin-yang codec is actually designed to obtain a balanced encoding, it has no error correction capability in theory.

4. although the authors make many simulation study, I want to know what the important insight can we get from your study?

5. To take a comprehensive comparison, the author should compare based on logical density, error capability, time complexity, cost etc to give us a general view of these method.

6. Fig 1c shows the tolerance of individual error type which I think is meaningless, as the combination of them is the reality and more challenging.

7. concerning the clustering algorithm, obviously, the quality of clustering have great influence on the recovering process of any codec.

(Remarks on code availability)

Reviewer #2

(Remarks to the Author)

The authors present a detailed comparative study of state-of-the-art codecs for DNA data storage under unified synthetic and experimental frameworks. The analysis focuses on the error-correction capabilities of existing codes, addressing both base-level errors and sequence dropouts. This has important implications for synthesis and sequencing costs, as well as storage density. The contribution is highly valuable to the DNA data storage community and provides clear benchmarks for future research. All claims made by the authors are well-supported by thorough evidence, and the methodology is clearly described. I do, however, have a number of concerns and general comments detailed below.

*Major comments:

- One of the main conclusions of the paper, as stated in the abstract "We find synthetic benchmarks commonly used in literature to be unsuitable indicators of codec performance..."; does not strike me as particularly surprising or intriguing,

given the simplistic methodology adopted for synthetic benchmarking. For example, the setup in Fig. 1(e) assumes exactly 30 reads for a subset of the strands and zero reads for the rest (controlled via dropout rate). It is evident that this “30 or zero” setup is far from realistic, and the resulting outcomes are not suitable indicators of real-world performance. In fact, prior work by the authors and others has shown that the coverage distribution is positively skewed and can be well-approximated by a log-normal distribution. Such a model could have been adopted for synthetic benchmarking, making the comparison between synthetic and real settings more meaningful. With relatively little effort, the synthetic setup could have been made more realistic, and any observed discrepancies between the synthetic and real cases would then have carried more weight. As it stands, I believe this is the main weakness of the study, and I provide further comments on this point in the sequel.

- In the “basic error scenario,” the meaning of “without clustering” is unclear. Does this imply that only a single reads of each strand is available for decoding? If multiple copies are available, are they ordered, indexed, or otherwise identifiable? If the decoder receives 30 noisy, unordered copies per oligo, it is unclear how decoding can proceed without a clustering step. Please clarify the assumptions and the exact decoding procedure in this scenario.

- While I understand that restricting the experiments to a single core eliminates the influence of parallelization and provides a uniform basis for comparison, this setting may also be unfair for certain codecs. Specifically, the “lack of parallelization in a codec’s implementation” is not always just an implementation limitation. In some cases, it reflects an inherent property of the decoding algorithm itself, which may not be amenable to parallel processing. This is a meaningful disadvantage of the codec that should not be overlooked in a comparative evaluation. I recommend clarifying and discussing this point.

- The imposed time constraint (5.4 bytes/second) should be justified. Is this threshold based on practical considerations, or is it arbitrary? Would the conclusions change significantly if the threshold were set to, say, 4 bytes/second? Presenting decoding performance as a function of runtime, rather than relying on a single cutoff (whose value may be difficult to justify), would offer a more informative and flexible comparison.

- Fig. 1 (d) shows the effect of clustering + MSA + consensus on error tolerance. This improvement is due to a reduction in the effective error rate that the codec must handle. For instance, when we say a codec tolerates $x\%$ errors with 30 reads, it is actually tolerating a residual error rate $y\%$ after consensus. However, the relationship between raw and residual error rates is not reported. I strongly recommend including these values, as they provide insight into how much of the burden is handled by the codec versus the consensus step.

- Given that cost is a central limitation in DNA storage, I was expecting a more thorough discussion of this aspect. Specifically, which codecs are best positioned to minimize cost per information bit while maintaining reliable decoding? For instance, the codec that allows the highest information density (bits/nt) under realistic error rates likely enables the lowest synthesis cost. Anchoring the comparison in terms of potential cost-per-bit would add significant value to the work.

- The paper frequently refers to “physical redundancy,” but its role in performance is not thoroughly discussed. If measured before amplification, and assuming relatively uniform physical coverage, why is physical redundancy critical? Is it because low physical redundancy can lead to synthesis errors being systematically propagated across all reads of a given strand? Or is it because low redundancy exacerbates coverage bias, increasing the likelihood of strand dropout? Clarifying this point would improve the discussion.

*Some minor comments:

- There is a missing punctuation mark in line 133.

- The terms “accuracy” and “similarity” appear to be used interchangeably in the main text and supplementary materials when describing clustering metrics. It would be helpful to adopt consistent terminology.

(Remarks on code availability)

I did not try running the code, but the provided Github repository seems to include all the necessary source files.

Reviewer #3

(Remarks to the Author)

The paper “Comparison of state-of-the-art error-correction coding for sequence-based DNA data storage” by Gimpel and colleagues is well written, and the objectives are clearly presented. The goal of the manuscript is to demonstrate how DNA storage algorithms (i.e., error-correcting codes) can be realistically analyzed and compared with respect to their performance for long-term storage. The study presents a benchmark of six representative DNA data storage codes, using both in silico simulations and in vitro experiments under standardized conditions.

However, the paper has several limitations that reduce the interpretability of its conclusions.

1. The selection of only six codecs: While justified by implementation availability and literature prominence, several recent codes are excluded, potentially limiting the claim of establishing the true state-of-the-art. Including a broader or more diverse codec set would provide a more comprehensive view of the field.

2. Simplified error models used for synthetic benchmarking: This is very similar to what has been done in the literature and raises the question of novelty. This approach, while standardized, potentially overlooks critical edge-case behaviors that

would be revealed under more diverse or adaptive error distributions, e.g., specific motifs that affect sequencing or synthesis.

3. Fixed parameter sets without optimization: The study relies on fixed parameter sets for each code without performing any code-specific parameter optimization. As a result, some codes may appear to underperform not due to intrinsic limitations but because their configurations were suboptimal for the tested conditions. This could be addressed by including a sensitivity analysis or an optimization run to assess how performance scales with parameter tuning.

4. Lack of computational performance analysis: A detailed analysis of runtime and memory usage is also required. A comprehensive benchmark should include not only error-correction capabilities but also computing performance measures. For real-world DNA storage systems, the speed and scalability of decoding pipelines are crucial performance metrics, and omitting them leaves a gap in the overall assessment.

5. Codec-clustering algorithm pairing: The pairing between codes and clustering algorithms significantly impacts performance. While the paper tests various clustering strategies and selects the best-performing one for each code, it does not fully explore whether these pairings are theoretically optimal or practically justified. In some cases, general-purpose tools like CD-HIT outperformed specialized algorithms, which raises questions about code-specific biases or incompatibilities with clustering strategies. A deeper investigation into why certain pairings fail, or how code design interacts with clustering, would have added depth to the analysis.

6. Importance of motifs in realistic evaluation: It is also worth mentioning that motifs are highly important for any realistic evaluation. There are known sequence motifs that interact with enzymes during sequencing and also some affecting DNA synthesis. PCR and other processes are also influenced by sequence motifs (or more precisely, by how these motifs interact with enzymes). This important fact should be more prominently highlighted, as only one code in the tested list can handle motifs. This directly relates to my first point: the selection of codes. There are other published codes that can handle sequence motifs. A fair, and more importantly, a realistic comparison would include motif-aware codes. A code that is not able to handle motifs will inevitably fail from time to time in practice, especially when large amounts of data are encoded.

In conclusion, the paper provides a valuable and ambitious framework for standardized code benchmarking in DNA data storage. However, its conclusions are somewhat constrained by the limited code selection, oversimplified error models, lack of parameter optimization, and absence of computational performance metrics. Expanding the scope and granularity of the evaluation would further enhance the robustness and applicability of the findings.

(Remarks on code availability)

Version 1:

Reviewer comments:

Reviewer #1

(Remarks to the Author)

The authors have solved most of my concerns, but I still concern the following issues:

1. I insist that the authors should select real SOTA codec to make comparison in the main text instead of adding some results in supplementary section.
2. I suspect the result in Supplementary Figure 21 (a) and (b) about the error tolerance rate about modulation based codec as it has a super error correction capability than any current codecs.
3. In fig 1 (d), concerning the error rate per nt, I want to know what the error types is, it means the sum of the insertion-deletion-substitution? further, what the ratio of the three errors? this is very important as I mention last time that in DNA storage, the unique error pattern is the complex IDS errors which makes the information retrieval is very challenge.
4. As this paper intend to make a comprehensive comparison of SOTA codes in error correction, it is very valuable to discuss which codec have the potential to deal with future large scale applications considering cost, time and error capability.

(Remarks on code availability)

Reviewer #2

(Remarks to the Author)

In my view, the authors have made a substantial effort to address the concerns raised in the first round of reviews. Given the large number of points raised, I will focus primarily on those I emphasized in my initial review. I find that these have been addressed to my satisfaction, and the quality of the paper has improved considerably compared to the original submission.

(Remarks on code availability)

Reviewer #3

(Remarks to the Author)

The authors addressed my concerns adequately.

(Remarks on code availability)

Version 2:

Reviewer comments:

Reviewer #1

(Remarks to the Author)

The authors have solved part of my concerns, but I still have some main concerns:

1. The YY code has no error correction ability, but your result shows it tolerates about 4% errors (Fig 1d). Could you explain the reason? Moreover, why did you select it as a SOTA method to study the error correction capability in this paper?
2. AI-based techniques have been introduced to solve the error issues in DNA storage, and recent studies showed that they have promising results both in error correction, data compression and encoding etc. But this paper neglects the function of these studies. As I know, compared with the traditional methods used in this paper, deep learning methods are more suitable for future DNA storage.
3. Concerning the modulation-based method, I agree it could not solve sequence loss, but HEDGES and DBGPS also can not deal with such issues. I still suspect the reported results in this paper.
4. Last, the loss issue in DNA storage is usually solved by fountain code or inner-outer code strategy. Therefore, base errors in sequences are usually corrected by various methods, which may combine the fountain code or inner-outer code strategy.

(Remarks on code availability)

Response to Referees

Reviewer comments in *italics*, author replies in **red** with actions **bolded**.

Line numbers refer to the revised manuscript, in which changes have been highlighted in yellow.

As a **general comment** we would like to state that it is/was not the goal of this work to identify one codec as being better than another. Rather, it was our goal to analyse trade-offs in existing coding strategies and ensure that future development of error correction codes for DNA data storage has a strong fundament to compare to. This is also nicely exemplified in the results, as current codes based on similar principles of inner and outer redundancy converge to a common state of the art (see Fig. 3 and new Fig. S21). We have adapted the text to make this intention more clear to the readers.

Comments by Referee #1

DNA storage is emerging as a potential alternative to the challenge of exponentially increasing data. One major topic is how to effectively retrieve information from the IDS error caused by DNA synthesis, PCR and sequencing process. and the dropout issue in the random sampling process is also a critical issue to be solved. Up to now, many methods have proposed to tackle these problems. this paper intended to make a comprehensive comparison of some famous codec to evaluate their performance. Although this work is very meaningful for us to understand the real performance of these methods, there exist some issues need to be solved:

We would like to thank the reviewer for critically assessing our manuscript and providing valuable comments.

1. the dropout issue is a unique problem in DNA storage in the random sampling process, it is relative easy to solve by the fountain code which is very mature in tradition communication society. And any codec could use it to solve this issue. therefore, the key issue should be the IDS error correction.

my suggestion is to separate the evaluation of IDS error correction capability and dropout issue.

We thank the reviewer for their suggestion of separating the evaluation of errors and sequence dropout. While we agree that fountain codes are mature, these also compete with other codes in traditional communications technologies and storage, including Reed Solomon codes (RAID-6 storage), BCH codes (SSD discs), Low Density Parity Check codes (Ethernet), and Turbo codes (3G/4G mobile communications). Consequently, it is not established that Fountain codes are the best solution for dropout errors specifically. Still, we agree that separating IDS errors and sequence dropout is interesting. Thus, following the reviewer's suggestion, we **have added new results on the tolerance to sequence dropout** as Fig. 1c. With this change, the separate evaluation of IDS error correction capability (via Fig. 1d) and sequence dropout (via Fig. 1c) is now possible. We have also **changed the corresponding section of the results section (II. 237-242) to highlight this distinction:**

“Thus, we extended the aforementioned basic error scenario with another variable, the fraction of sequences lost. As such, Fig. 1c shows the rate of sequence loss each codec tolerates in the absence of errors. The large differences in tolerated sequence loss between codecs and code rates – from 0% (Goldman and Yin-Yang) to 64% (DNA Fountain at 0.5 bit/nt) – highlight that codecs are not equally equipped to compensate for this unique problem in DNA data storage.”

a End-to-end overview of the DNA data storage pipeline

b Estimating error tolerance for successful decoding

c Tolerances against sequence loss

d Effect of clustering on error tolerances of codecs

e Pareto fronts between errors and sequence dropout

Figure 1: Overview of the scope of this work and the evaluations of clustering algorithms and codecs. (a) Overview of the data storage workflow considered in this work, including encoding of the data with a codec, an *in silico* or *in vitro* data storage workflow, post-processing by read clustering and generation of consensus sequences, and decoding of the data with a codec. (b) Exemplary outcome of a basic error scenario with naïve (grey) or CD-HIT clustering (red) upon variation of the overall error rate (see Methods). Individual points denote the outcomes of 30 individual iterations of the scenario. The solid lines represent the logistic regression performed to estimate the error rate at which data recovery succeeds with 95% probability (dashed grey line). The corresponding error rate is then used as performance metric in this work. (c) Tolerance of all codecs in the basic error scenario to sequence loss. All codecs use the best-performing clustering algorithm denoted in panel d. Conditions indicated with an hourglass (X) were limited by the time constraint (see Supplementary Figs. 6-7). (d) Performance of each codec at its supported code rates in the basic error scenario, using both naïve clustering (grey bars) and the best-performing clustering algorithm for each codec (colored bars, see Supplementary Table 3 for full data). Conditions indicated with an hourglass (X) were limited by the time constraint (see Supplementary Figs. 4-5). Dark shading of the bars indicates the effective error rate remaining after the clustering step. (e) Pareto fronts of codec performance in a scenario combining errors at a fixed ratio of 53% substitutions, 45% deletions, and 2% insertions with sequence dropout. Feasible regions lie below the indicated pareto fronts, with the mean error rates of the high-fidelity (dashed line) and low-fidelity scenario (dotted line) indicated. The Yin-Yang codec, which did not include independent error-correction capabilities, was unable to decode the data in this scenario at all, given that neither it nor the clustering step was capable of compensating for any sequence dropout.

(conical microtube, pipette, and thermocycler) were provided by Labicons (www.labicons.net). Panel a) partially created in BioRender. Gimpel, A. (2025) <https://BioRender.com/9wrsaez>.

2. what is the criterion of the six SOTA methods? I suggest the author should select more representative methods:

(1) The modulation-based codec which can tolerate high IDS errors (<https://pubs.acs.org/doi/abs/10.1021/acs.jcim.3c00629>)

(2) deep learning based methods (<https://onlinelibrary.wiley.com/doi/abs/10.1002/smt.202400959>, <https://www.nature.com/articles/s41598-024-83806-5>)

As described in the methods of the manuscript, we selected the six codecs based on “availability of an open-source implementation with sufficient documentation, the presence of in vitro experiments in the original publication, the prominence in literature, and considerations for covering a broad range of approaches”, with October 2023 as the cut-off date (due to lead time for experimental work). We **extended our description of selection criteria at the beginning of the results section (ll. 109-112)** to better document our selection:

“For this study, we limited our benchmarking of codec performance to representative examples, using the availability of an open-source implementation with sufficient documentation, the presence of in vitro experiments in the original publication, the prominence in literature, and considerations for covering a broad range of approaches as selection criteria (see Methods and Supplementary Table 2, cut-off date October 2023).”

The first method suggested by the reviewer, from Zan et al.¹, was initially excluded because its study did not feature any vitro experiments (see Supplementary Table 2). Both deep learning-based methods^{2,3} suggested by the reviewer were published after our cut-off date (both in December 2024) and do not provide an open-source code repository. Nonetheless, we **have now added new results comparing the performance of the state-of-the-art with additional literature methods, specifically those by Zan et al. (Modulation)¹ proposed by the reviewer, as well as Chandak et al. (LDPC)⁴, and Song et al. (DBGPS)⁵**. The results are shown in Supplementary Fig. 21 and discussed in ll. 404-412 of the results:

“To demonstrate the versatility of the established high- and low-fidelity scenarios for codec benchmarking, we extended our benchmarking with three additional codecs from the literature, namely Modulation by Zan et al.¹, LDPC by Chandak et al.⁴, and DBGPS by Song et al.⁵ Supplementary Fig. 21 shows the performance of these additional codecs compared to the best-performing codecs from Fig. 3, DNA-Aeon by Welzel et al.⁶ and DNA-RS by Heckel et al.^{7,8} Interestingly, DBGPS by Song et al.⁵ achieved a similar performance to DNA-Aeon and DNA-RS in most scenarios. This highlights the value of standardized benchmarks for codec comparisons and suggests using a balanced inner/outer code separation strategy is more important than choosing a specific error-correction code and implementation.”

Supplementary Figure 21: Extension of the in-silico analysis to additional literature codecs by Song et al. (DBGPS)⁵, Chandak et al. (LDPC)⁴ and Zan et al. (Modulation)¹. (a) The effect of clustering on the additional literature codecs, as measured in tolerated error rate, between naïve clustering (grey) and the best-performing clustering algorithm (green, CD-HIT). (b) Comparison of each codec’s error tolerance against individual error types (deletions, blue; insertions, green; substitutions, orange). The best-performing codecs from the main text, DNA-Aeon by Welzel et al.⁶ and DNA-RS by Heckel et al.^{7,8} are shown for comparison. (c) Pareto fronts of the additional codecs upon simultaneous consideration of errors and sequence dropout, grouped by code rate. The best-performing codecs from the main text, DNA-Aeon by Welzel et al.⁶ and DNA-RS by Heckel et al.^{7,8} are shown in lighter colours for comparison. (d) Pareto fronts of the additional codecs in the high- (top row) and low-fidelity scenarios (bottom row) upon variation of the initial coverage and the sequencing depth. In the high-fidelity scenario, synthesis by material deposition and amplification with a high-fidelity polymerase is assumed, whereas the low-fidelity scenario includes electrochemical synthesis and amplification with Taq polymerase. The best-performing codecs from the main text, DNA-Aeon by Welzel et al.⁶ and DNA-RS by Heckel et al.^{7,8} are shown in lighter colours for comparison.

3. the Yin-yang codec is actually designed to obtain a balanced encoding, it has no error correction capability in theory.

We fully agree with the reviewer. The Yin-Yang codec was included in our analysis to showcase the possibility of error-free data decoding without any codec-level error-correction. We have **highlighted the lack of error-correction capability for the Yin-Yang codec at multiple locations throughout the manuscript** (e.g., ll. 105, 186, 215, 344, 567, Table 1).

4. although the authors make many simulation study, I want to know what the important insight can we get from your study?

We thank the reviewer for the opportunity to improve the description of the main outcomes of our study. We **rephrased the abstract (ll. 23-36) and the conclusions (ll. 493-494, 498-501, 507-513)** as follows:

“[...]. Using synthetic benchmarks, we find that existing codecs can tolerate error rates up to 14% and sequence loss as high as 65% in isolation. However, codecs weigh redundancy to errors and sequence loss differently, calling the simple metrics commonly used for assessing codec performance in literature into question. To improve upon these simple metrics, we implement comprehensive error scenarios covering major experimental parameters to assess codec performance under realistic conditions. In testing codecs using both high- and low-fidelity workflows, a balanced inner/outer code separation strategy emerges as superior error-correction scheme, independently of the implementation. Moreover, storage densities as high as 117 EB g⁻¹ are feasible using existing codecs and current synthesis and sequencing technologies. Verifying our results with fair, standardized in vitro experiments, we demonstrate data storage at 43 EB g⁻¹ using synthesis by material deposition and 13 EB g⁻¹ using the more error prone electrochemical synthesis, employing only existing codecs from the literature. Besides closing in on the physical limits of DNA data storage, this study thus demonstrates the maturity of error-correction coding, defines its current state-of-the-art, and establishes best practices for codec benchmarking.”

“Notably, we find that existing codecs can tolerate error rates up to 14% and sequence loss as high as 65% in isolation, but codecs’ abilities to correct both simultaneously varies drastically. [...]. Taken together, our results thus challenge common practice in the DNA data storage literature, as both synthetic error benchmarks and serial dilution/amplification experiments are widely-used standards to showcase codec performance,^{6,9-11} but fail to differentiate codec performance in our standardized setup. As an alternative, rigorous benchmark, we propose and implement both a high- and low-fidelity scenario, thereby systematically assessing codecs across three critical experimental factors: synthesis technology, physical redundancy, and sequencing depth. [...]. Extension of our synthetic benchmark suite to three additional literature codecs (see Supplementary Fig. 21) further highlighted that state-of-the-art performance depended more on a well-executed, balanced inner/outer code separation strategy than on the specific choice of error-correction code. This suggests existing codecs are sufficiently close to the theoretical limit, such that minor implementation details (e.g., optimality of parameter choices, balance of inner and outer redundancy) cause the remaining, minor gaps in codec performance in most cases.”

5. To take a comprehensive comparison, the author should compare based on logical density, error capability, time complexity, cost etc to give us a general view of these method.

We appreciate the reviewer’s suggestion regarding important aspects of a comprehensive comparison. We’d like to point out that our study now includes all of the mentioned aspects:

- we systematically adjust codec parameters to achieve three standardized logical densities (referred to as code rates in the manuscript) of 0.5, 1.0, and 1.5 bit/nt where possible,
- we quantify error capability with both simple error models for error and dropout tolerance (see Fig. 1e), as well as realistic error models which additionally address sequencing depth and physical coverage (see Fig. 3a),

- we include time complexity in our results by imposing a compute and time constraint, thereby penalizing poorly scaling methods (see e.g., Supplementary Figs. 4-5),
- we have now also **added a cost comparison using the cost structures of both Twist and Genscript** as Supplementary Fig. 20.

6. *Fig 1c shows the tolerance of individual error type which I think is meaningless, as the combination of them is the reality and more challenging.*

We thank the reviewer for their suggestion. We have now **replaced the panel in Fig. 1c, showing the tolerance to individual error types, with a panel showing the tolerance to sequence dropout**, as suggested in this reviewer's first comment. The tolerance to individual error types is now shown as Supplementary Fig. 8 instead. We have **amended the main text to reflect this change** (ll. 237-242).

7. *concerning the clustering algorithm, obviously, the quality of clustering have great influence on the recovering process of any codec.*

We agree with the reviewer's assessment of the relevance of clustering quality on decoding performance. For this reason, we tested all selected codecs with five different clustering algorithms (see Supplementary Tables 1 and 3) before selecting the best-performing clustering algorithm for each codec. Indeed, as proposed by the reviewer, the choice of clustering algorithm had remarkable impact on the codec performance.

As Reviewer 3 also suggested a detailed investigation into the impact of the selected codec-clustering pairings, we **performed additional analyses to quantify the reduction in effective error rate for all codec-clustering pairings**, see Supplementary Figs. 1 and 3. We **added a paragraph outlining the findings to the results section** (see ll. 181-183):

"As expected, this positive effect on error tolerance was caused by a drastic reduction in the error rate within the clustered reads (as low as 0% up to an error rate of 5%, see Supplementary Figs. 1 and 2), thereby considerably reducing the effective error rate for the decoder."

Supplementary Figure 1: Relationship between the effective error rate in clustered reads and the applied error rate during the basic error scenario for each codec-clustering pairing. For each investigated codec (rows), code rate (columns), and clustering algorithm (color), the effective error rate in the clustered reads as determined by mapping to the reference sequences is shown. In all cases, an increase in the error rate in the sequencing reads yields an increase in the error rate in the clustered reads.

Comments by Referee #2

The authors present a detailed comparative study of state-of-the-art codecs for DNA data storage under unified synthetic and experimental frameworks. The analysis focuses on the error-correction capabilities of existing codes, addressing both base-level errors and sequence dropouts. This has important implications for synthesis and sequencing costs, as well as storage density. The contribution is highly valuable to the DNA data storage community and provides clear benchmarks for future research. All claims made by the authors are well-supported by thorough evidence, and the methodology is clearly described. I do, however, have a number of concerns and general comments detailed below.

We thank the reviewer for their positive evaluation of our work and their great suggestions.

**Major comments:*

- One of the main conclusions of the paper, as stated in the abstract “We find synthetic benchmarks commonly used in literature to be unsuitable indicators of codec performance...”; does not strike me as particularly surprising or intriguing, given the simplistic methodology adopted for synthetic benchmarking. For example, the setup in Fig. 1(e) assumes exactly 30 reads for a subset of the strands and zero reads for the rest (controlled via dropout rate). It is evident that this “30 or zero” setup is far from realistic, and the resulting outcomes are not suitable indicators of real-world performance. In fact, prior work by the authors and others has shown that the coverage distribution is positively skewed and can be well-approximated by a log-normal distribution. Such a model could have been adopted for synthetic benchmarking, making the comparison between synthetic and real settings more meaningful. With relatively little effort, the synthetic setup could have been made more realistic, and any observed discrepancies between the synthetic and real cases would then have carried more weight. As it stands, I believe this is the main weakness of the study, and I provide further comments on this point in the sequel.

We thank the reviewer for pointing out the perceived simplicity of our synthetic benchmarks. We'd like to clarify that we use two different error models for our synthetic benchmarks throughout our manuscript, precisely to support our conclusion regarding the inadequacy of simplistic error models. The simple model referred to by the reviewer (i.e., exactly 30 reads with fixed error rates and sequence dropout) represents the status-quo in the literature, and is used only for the initial analyses in Fig. 1. The subsequent analyses in Figs. 2 and 3 use a complex error model based on exactly the previous work¹² by us the reviewer mentioned. This model includes:

- skewed coverage distributions, as highlighted by the reviewer,
- sequence-specific efficiency bias during PCR amplification,
- consecutive deletions (“deletion runs”) from synthesis,
- positional dependence of the error rate during sequencing,

among other non-idealities common in real-world sequencing data.

We'd also like to point out our experimental confirmation of the results obtained with the complex model (see Fig. 4). The good agreement between experiments and our model results highlights how realistically our complex model can represent experimental sequencing data.

To better differentiate the two models used throughout the study and to highlight the complexity of our synthetic benchmarks, we **rephrased the corresponding paragraphs in the introduction and results** (ll. 89-93 and ll. 274-280):

“For our *in silico* benchmarks, we first employ simple synthetic error scenarios (i.e., applying fixed error rates) commonly used in the literature. Then, we extend our analysis with comprehensive benchmarks considering the full spectrum of non-idealities present in real-world sequencing data (using DT4DDS¹²), and verify our results with standardized *in vitro* experiments”

“Despite the common use of these simple synthetic benchmarks for codec benchmarking in the literature, they overlook the non-ideal error patterns and biases present in experimental sequencing data – from error runs to skewed coverage distributions.¹²⁻¹⁴ Thus, for all further benchmarks, we implemented realistic experimental workflows using the models implemented in the simulation software DT4DDS¹² (see Methods). This simulation software takes into account the aforementioned non-idealities which are missing from the synthetic benchmarks previously performed in the literature.”

- In the “basic error scenario,” the meaning of “without clustering” is unclear. Does this imply that only a single reads of each strand is available for decoding? If multiple copies are available, are they ordered, indexed, or otherwise identifiable? If the decoder receives 30 noisy, unordered copies per oligo, it is unclear how decoding can proceed without a clustering step. Please clarify the assumptions and the exact decoding procedure in this scenario.

We appreciate the reviewer highlighting this unclear point. “Without clustering” refers to the situation in which the set of all 30 reads per sequence are supplied to the decoder (i.e., all unique reads). This means the decoder has multiple erroneous, unordered copies available for decoding. All encoders investigated in this study include indexes in their sequences, so that decoding can proceed without requiring the ordering of copies during a separate clustering step. We have **improved the explanation of the assumptions and the decoding procedure in this situation** (ll. 151-153):

“Fig. 1d shows the error tolerance of all codecs in this basic error scenario, without clustering and consensus generation by alignment (i.e., the set of all 30 erroneous reads per sequence are supplied to the decoder directly, following the procedure by Erlich et al.⁹; shown as “Naïve”, grey bars).”

- While I understand that restricting the experiments to a single core eliminates the influence of parallelization and provides a uniform basis for comparison, this setting may also be unfair for certain codecs. Specifically, the “lack of parallelization in a codec’s implementation” is not always just an implementation limitation. In some cases, it reflects an inherent property of the decoding algorithm itself, which may not be amenable to parallel processing. This is a meaningful disadvantage of the codec that should not be overlooked in a comparative evaluation. I recommend clarifying and discussing this point.

We agree with the reviewer’s point about potentially disadvantaging parallelized codecs by constraining codecs to one core. For our evaluation, we preferred to compare codecs using comparable compute power, rather than to introduce the additional dimension of parallelizability. As noted by the reviewer, some codecs might not be parallelized due to inherent limitations of the algorithm, while other codecs might be parallelizable in theory but weren’t implemented as such. We decided that enabling parallelization would misrepresent the latter group, while skewing the overall results based on the larger amounts of compute power available to parallelized codecs.

We now **clarify our decision and discuss this point early in the results** (ll. 147-150):

“Moreover, the constraint to one CPU core negated any undue disadvantage due to a lack of parallelization in a codec’s implementation. However, this constraint thus overlooks inherent incompatibilities with parallel processing possibly present in some decoding algorithms, which represent a considerable disadvantage to their large-scale implementation.”

- The imposed time constraint (5.4 bytes/second) should be justified. Is this threshold based on practical considerations, or is it arbitrary? Would the conclusions change significantly if the threshold were set to, say, 4 bytes/second? Presenting decoding performance as a function of runtime, rather than relying on a single cutoff (whose value may be difficult to justify), would offer a more informative and flexible comparison.

The reviewer raises an important point about the impact of our time constraint on our results. We chose the time constraint based on practical considerations for our computing environment, rather than based on anticipated decoding throughput. Specifically, the time constraint of one hour per decoding attempt ensures that all decoding attempts can finish within the time limit for a single computing job on our computing cluster. The specified decoding throughput of 5.4 bytes/second thus arises indirectly from the time limit (1h) and the file size (19 kB). While the value of this minimum throughput is thus arbitrary, it is sufficiently low such that any codec which is limited by this constraint would be unsuitable for most applications of DNA data storage (e.g., cold storage of large archives).

We show the decoding performance relative to the runtime in Supplementary Figs. 4, 6, 9, 11, 12, and 13. Evidently, the time constraint is not the limiting factor in the vast majority of cases. In the few cases in which time is constraining codec performance, the dependence on the exact time limit is minor. We now **discuss the dependence of decoding performance on our choice of time limit in the results** (ll. 140-146):

“The choice of time constraint and file size were based on limitations of the employed computing environment, and thus indirectly enforced a decoding speed of at least 5.4 bytes per second. While this minimum enforced decoding throughput was thus not based on practical considerations, it is nonetheless sufficiently permissive such that any codec which is limited by this constraint would be unsuitable for most applications of DNA data storage (e.g., cold storage of large archives). Accordingly, the time constraint did not limit decoding in the vast majority of cases, and we show the decoding performance relative to the runtime in Supplementary Figs. 4, 6, 9, 11, 12, and 13.”

- Fig. 1 (d) shows the effect of clustering + MSA + consensus on error tolerance. This improvement is due to a reduction in the effective error rate that the codec must handle. For instance, when we say a codec tolerates x% errors with 30 reads, it is actually tolerating a residual error rate y% after consensus. However, the relationship between raw and residual error rates is not reported. I strongly recommend including these values, as they provide insight into how much of the burden is handled by the codec versus the consensus step.

We agree that analysing the reduction in effective error rate afforded by clustering is an important aspect of understanding the effects of clustering on decoding performance. We have **performed a new, in-depth investigation into the effects on error rates for all codec-clustering pairings** and report them in Supplementary Fig. 2. Moreover, we have **revised Fig. 1d in the main text to show the effective error rate after clustering to facilitate a direct comparison.**

To highlight this effect of clustering, we **included more details in the corresponding two paragraphs of the results section** (see ll. 181-183, 217-221):

“As expected, this positive effect on error tolerance was caused by a drastic reduction in the error rate within the clustered reads (as low as 0% up to an error rate of 5%, see Supplementary Figs. 1 and 2), thereby considerably reducing the effective error rate for the decoder. [...]. Interestingly, nine of the thirteen tested codecs and code rates performed best with the established general-purpose clustering algorithm CD-HIT (red bars in Fig. 1d). Unsurprisingly, this general preference for CD-HIT was caused by this clusterer’s superior performance, yielding the fewest and least erroneous clusters across all tested clustering algorithms up to an overall error rate of around 10% (see Supplementary Figs. 1-3).”

Supplementary Figure 2: Relationship between the effective error rate in the closest clustered reads and the applied error rate during the basic error scenario for each codec-clustering pairing. For each investigated codec (rows), code rate (columns), and clustering algorithm (color), the effective error rate in the clustered reads as determined by mapping to the reference sequences is shown. In contrast to Suppl. Fig. 1, only the closest cluster to each reference sequence is considered. This is most representative of the effective error rate encountered by the decoder. In all cases, an increase in the error rate in the sequencing reads yields an increase in the error rate in the clustered reads.

- Given that cost is a central limitation in DNA storage, I was expecting a more thorough discussion of this aspect. Specifically, which codecs are best positioned to minimize cost per information bit while maintaining reliable decoding? For instance, the codec that allows the highest information density

(bits/nt) under realistic error rates likely enables the lowest synthesis cost. Anchoring the comparison in terms of potential cost-per-bit would add significant value to the work.

We thank the reviewer for their great suggestion regarding the comparison of synthesis costs. Importantly, the cost structures of commercial synthesis providers are not linear, meaning the cost of synthesizing one base decreases drastically as the overall size of the library grows (see Supplementary Fig. 20a+b). As a result, the cost per information bit depends much more strongly on the data volume than on the codec choice.

With this in mind, we have **added a comparison using the cost structures of both Genscript and Twist, comparing the cost per information bit for storing one megabyte of data using all investigated codecs and code rates** (see Supplementary Fig. 20c). As predicted by the reviewer, the codecs with the highest information density (i.e., 1.5 bit/nt) have the lowest synthesis cost per bit of information.

We now also highlight these findings in the results (ll. 394-403):

“However, choosing lower code rates increases synthesis costs proportionally, as the price of synthesis scales directly with the number of sequences to be synthesized resp. the amount of data stored (see Supplementary Fig. 20a+b). Given the standardized code rates used in this study, all codecs feature similar costs per bit stored, at 14 USD/kB on average in the high-fidelity scenario and 7.9 USD/kB in the low-fidelity scenario (at a 1 MB scale, see Supplementary Fig. 20c). Ultimately, synthesis cost is not directly defined by the choice of codec, but purely by the employed code rate. To this end, the choice of code rate must ultimately compromise between achieving the highest storage densities at lower code rates and reducing the synthesis costs at higher code rates. Thus, given the direct scaling of synthesis costs with code rate, our performance analysis comparing three standardized code rates also translates directly to anticipated synthesis costs.”

- The paper frequently refers to “physical redundancy,” but its role in performance is not thoroughly discussed. If measured before amplification, and assuming relatively uniform physical coverage, why is physical redundancy critical? Is it because low physical redundancy can lead to synthesis errors being systematically propagated across all reads of a given strand? Or is it because low redundancy exacerbates coverage bias, increasing the likelihood of strand dropout? Clarifying this point would improve the discussion.

We define physical redundancy as the mean number of copies per reference sequence during storage (i.e., after synthesis, amplification, and dilution; but before re-amplification and sequencing). As correctly identified by the reviewer, physical redundancy is critical as an experimental parameter because a low physical redundancy increases the risk that sequences are missing in the sequencing data, and because the remaining reads feature similar errors (thereby reducing the effectiveness of clustering). Importantly, physical redundancy is also an important metric in itself, as it is inversely proportional to storage density, one of DNA’s main advantages as a storage medium.

We have now extended our discussion of physical redundancy as an experimental parameter, highlighting its effects on sequencing output (ll. 321-328):

“While the choice of synthesis provider is usually well-documented in literature studies and its effect on error rate well-established, the choices of physical redundancy and sequencing depth vary considerably across studies and affect only the homogeneity of sequencing reads.^{12,13,15}

Specifically, both low physical redundancies and low sequencing depths increase the likelihood of sequence dropout and diminish the diversity of the oligos due to the stochastic sampling of only few oligo copies. Importantly, physical redundancy directly affects the data's storage density (i.e., it is inversely proportional to storage density), whereas sequencing depth directly influences sequencing costs."

a Price per base by sequence count and provider

b Price per kilobyte by data volume and provider

c Price per kilobyte by codec and scenario at the 1 MB scale

Supplementary Figure 20: Cost comparison of codecs by synthesis provider and scenario. (a) The cost structure of the two commercial providers analysed in this study depends strongly on the number of synthesized sequences (assuming a fixed sequence length of 150 nt). (b) Using a fixed code rate of 1.5 bit/nt for synthesis by Twist Biosciences and 1.0 bit/nt for Genscript yields comparable prices per kilobyte by data volume. (c) Comparing the costs for synthesis across all codecs and code rates tested in this study, in both the high- (blue) and low-fidelity scenarios (orange) and at a fixed data volume of one megabyte, reveals only minor differences. However, the costs for low-fidelity synthesis are about half as high as the high-fidelity synthesis across most codecs.

**Some minor comments:*

- There is a missing punctuation mark in line 133.

Thanks for pointing out this mistake. This has been fixed.

- The terms “accuracy” and “similarity” appear to be used interchangeably in the main text and supplementary materials when describing clustering metrics. It would be helpful to adopt consistent terminology.

We thank the reviewer for bringing up this inconsistency. **The use of these terms has now been harmonized.**

Reviewer #2 (Remarks on code availability):

I did not try running the code, but the provided Github repository seems to include all the necessary source files.

We thank the reviewer for evaluating the completeness of our code repository.

Comments by Referee #3

The paper "Comparison of state-of-the-art error-correction coding for sequence-based DNA data storage" by Gimpel and colleagues is well written, and the objectives are clearly presented. The goal of the manuscript is to demonstrate how DNA storage algorithms (i.e., error-correcting codes) can be realistically analyzed and compared with respect to their performance for long-term storage. The study presents a benchmark of six representative DNA data storage codes, using both in silico simulations and in vitro experiments under standardized conditions.

We appreciate the reviewer's positive evaluation of our manuscript and its goals. We also thank the reviewer for their constructive comments.

However, the paper has several limitations that reduce the interpretability of its conclusions.

1. The selection of only six codecs: While justified by implementation availability and literature prominence, several recent codes are excluded, potentially limiting the claim of establishing the true state-of-the-art. Including a broader or more diverse codec set would provide a more comprehensive view of the field.

We agree with the reviewer, and have now extended our analysis to three additional codecs from the literature. While we would have liked to use even more codecs from the literature, we observed that also many recent codecs are either lacking an open-source implementation altogether, or available implementations are poorly reusable, thus requiring undue reprogramming to allow for benchmarking. Moreover, many recent studies focus on image storage and compression, and are thus beyond the scope of our work on general-purpose methods.

Nonetheless, we **have now added new results comparing the performance of the state-of-the-art with additional literature methods, specifically those by Zan et al. (Modulation)¹, Chandak et al. (LDPC)⁴, and Song et al. (DBGPS)⁵**. The results are shown in Supplementary Fig. 21 and discussed in II. 404-412 of the results:

"To demonstrate the versatility of the established high- and low-fidelity scenarios for codec benchmarking, we extended our benchmarking with three additional codecs from the literature, namely Modulation by Zan et al.¹, LDPC by Chandak et al.⁴, and DBGPS by Song et al.⁵ Supplementary Fig. 21 shows the performance of these additional codecs compared to the best-performing codecs from Fig. 3, DNA-Aeon by Welzel et al.⁶ and DNA-RS by Heckel et al.^{7,8} Interestingly, DBGPS by Song et al.⁵ achieved a similar performance to DNA-Aeon and DNA-RS in most scenarios. This highlights the value of standardized benchmarks for codec comparisons and suggests using a balanced inner/outer code separation strategy is more important than choosing a specific error-correction code and implementation."

2. Simplified error models used for synthetic benchmarking: This is very similar to what has been done in the literature and raises the question of novelty. This approach, while standardized, potentially overlooks critical edge-case behaviors that would be revealed under more diverse or adaptive error distributions, e.g., specific motifs that affect sequencing or synthesis.

Reviewer 2 also remarked on the perceived simplicity of our synthetic benchmarks in their first comment above. As discussed in more detail in our response there, we'd like to clarify that we use two different error models for our synthetic benchmarks. A simple model, as commonly used in literature

(i.e., fixed error rates and sequence dropout), is used only for the initial analyses in Fig. 1. The subsequent analyses in Figs. 2 and 3 use a complex error model based on our previous comprehensive error characterization in Ref. 12. This includes diverse and complex error patterns which have not been used previously for codec comparisons, such as error runs, skewed coverage distributions, PCR biases, and positional biases.

Moreover, we have verified the results obtained from the complex error model with experiments, showing good agreement. This highlights how closely our complex model represents experimental sequencing data. This confirmation of benchmark results using experiments covering multiple codecs is also novel in itself.

To better highlight the complexity of our synthetic benchmarks and underline the novelty in their use for codec benchmarking, we **rephrased the corresponding paragraphs in the introduction and results** (ll. 89-93 and ll. 274-280):

“For our *in silico* benchmarks, we first employ simple synthetic error scenarios (i.e., applying fixed error rates) commonly used in the literature. Then, we extend our analysis with comprehensive benchmarks considering the full spectrum of non-idealities present in real-world sequencing data (using DT4DDS¹²), and verify our results with standardized *in vitro* experiments”

“Despite the common use of these simple synthetic benchmarks for codec benchmarking in the literature, they overlook the non-ideal error patterns and biases present in experimental sequencing data – from error runs to skewed coverage distributions.¹²⁻¹⁴ Thus, for all further benchmarks, we implemented realistic experimental workflows using the models implemented in the simulation software DT4DDS¹² (see Methods). This simulation software takes into account the aforementioned non-idealities which are missing from the synthetic benchmarks previously performed in the literature.”

3. Fixed parameter sets without optimization: The study relies on fixed parameter sets for each code without performing any code-specific parameter optimization. As a result, some codes may appear to underperform not due to intrinsic limitations but because their configurations were suboptimal for the tested conditions. This could be addressed by including a sensitivity analysis or an optimization run to assess how performance scales with parameter tuning.

The reviewer raises an important point about the choice of codec parameters throughout our study. We agree with the reviewer that the lack of parameter optimization may have led to the use of suboptimal parameter sets for some codec-scenario combinations, which we also acknowledged in our original discussion. However, our goal with this study is to compare existing codecs from an end-user perspective. Following this perspective, we always start from each codecs' default settings and refrain from extensive parameter optimization, which a common end-user would not have the ability to perform due to a lack of a priori knowledge of the error rates encountered in an experiment (i.e., usually codec parameters are estimated and not optimized, as there is no opportunity to test codec parameters prior to the experiment).

Moreover, we assume the original studies' authors have performed an appropriate selection of their codecs' parameters for common DNA data storage scenarios when publishing their study. Our changes to these parameters to achieve the standardized code rates and sequence lengths thus only change the overall redundancy level, while leaving other parameters, such as the balance between inner and outer redundancy, at approximately the values selected by the original authors.

To better highlight this intention and explain the rationale of using the codec settings as selected by the original authors as baselines for our parameter choices, we **added a paragraph to the beginning of the results** (see ll. 119-127):

“Importantly, codec parameters were not optimized specifically for this study. Instead, the parameters selected by the codecs’ original authors were only adjusted to achieve standardized code rates and similar sequence lengths. This effectively changed only each codec’s overall redundancy level (via the code rate), whereas other codec features – such as the balance between inner and outer redundancy – were kept at approximately the level selected by the original authors. While this approach might lead to suboptimal codec configurations in our benchmarks, this approach mimics the use case for experimentalists and matches each codec’s authors’ selected implementation most closely. This was deemed important, as the codecs’ original authors were assumed to have estimated their ideal parameters based on their specific use cases in DNA data storage.”

4. Lack of computational performance analysis: A detailed analysis of runtime and memory usage is also required. A comprehensive benchmark should include not only error-correction capabilities but also computing performance measures. For real-world DNA storage systems, the speed and scalability of decoding pipelines are crucial performance metrics, and omitting them leaves a gap in the overall assessment.

We agree with the reviewer’s assessment regarding the importance of performance measures for a comprehensive benchmark. This is the reason why we already enforced both runtime (60 min), memory (8 GB RAM), and computing (1 CPU core) constraints during all of our experiments. Supplementary Figures 4-7 and 9-13 **show the dependence of both runtime and memory usage on the investigated experimental parameters** (error rate, sequence loss, sequencing depth, physical redundancy). We have now also **added a comparison of the computational performance to the results section** (see ll. 259-263, 380-381) and **include an overview in the discussion** (see ll. 534-540), both reproduced below:

“The runtime constraint also limited codec performance of DNA-Aeon, DNA-RS, and HEDGES in this scenario (see Supplementary Fig. 11). Interestingly, both DNA-Aeon and HEDGES were only limited by time at high error rates while operating unconstrained at high sequence dropout, suggesting their implementations of within-sequence redundancy caused excessive runtimes. As previously, memory use did not limit any codec in this scenario. [...]. This suggests that also in this benchmark, high error-correction capability is in a direct trade-off with computational power. Expectedly, memory use did not limit codec performance in either scenario.”

“The analysis of the computational performance throughout all benchmarks in this study (see Supplementary Figs. 4-7 and 9-13) shows that only DNA-Aeon, DNA-RS, and HEDGES were limited by the runtime constraint at all. However, runtime generally increased sharply as error rates approached each codec’s limits, such that, in many cases, codecs failed to successfully decode the data before being limited by our time constraint. Nonetheless, codecs’ runtime and parallelizability should become an important metric for consideration as data volumes for DNA data storage continue to increase.”

a Runtime for DNA-Aeon**b Runtime for DNA Fountain****c Runtime for DNA-RS****d Runtime for Goldman****e Runtime for HEDGES****f Runtime for Yin-Yang**
■ Substitutions ■ Insertions ■ Deletions ● Decoding successful ○ Decoding unsuccessful

Supplementary Figure 9: Decoding time as a function of error rate in the basic error scenario with individual errors. The runtime of the decoding step is shown for the DNA-Aeon (a), DNA Fountain (b), DNA-RS (c), Goldman (d), HEDGES (e), and Yin-Yang (f) codecs at all used code rates, when substitutions (red), deletions (blue), or insertions (green) are introduced individually. Points correspond to individual runs of the pipeline at the specified error rate and error type. Open circles denote individual runs which failed the decoding step, either due to violation of the runtime constraint or due to insufficient error-correction capabilities.

a Memory use for DNA-Aeon**b** Memory use for DNA Fountain**c** Memory use for DNA-RS**d** Memory use for Goldman**e** Memory use for HEDGES**f** Memory use for Yin-Yang
■ Substitutions
 ■ Insertions
 ■ Deletions
 ● Decoding successful
 ○ Decoding unsuccessful

Supplementary Figure 10: Memory use as a function of error rate in the basic error scenario with individual errors. The memory use of the decoding step is shown for the DNA-Aeon (a), DNA Fountain (b), DNA-RS (c), Goldman (d), HEDGES (e), and Yin-Yang (f) codecs at all used code rates, when substitutions (red), deletions (blue), or insertions (green) are introduced individually. Points correspond to individual runs of the pipeline at the specified error rate and error type. Open circles denote individual runs which failed the decoding step, either due to violation of the runtime constraint or due to insufficient error-correction capabilities.

5. *Codec–clustering algorithm pairing: The pairing between codes and clustering algorithms significantly impacts performance. While the paper tests various clustering strategies and selects the best-performing one for each code, it does not fully explore whether these pairings are theoretically optimal or practically justified. In some cases, general-purpose tools like CD-HIT outperformed specialized algorithms, which raises questions about code-specific biases or incompatibilities with clustering strategies. A deeper investigation into why certain pairings fail, or how code design interacts with clustering, would have added depth to the analysis.*

We support the reviewer’s comments regarding the importance of the codec-clustering pairing. While we had already mentioned the possibility of certain code-specific biases as the reason for the low performance of some codec-clustering pairings, we have **performed a deeper investigation into these biases** as Supplementary Figs. 1-3 and Supplementary Table 4. As suspected by the reviewer, certain pairings fail due to the interaction between a codec’s sequence design and some clustering algorithm’s aggressive merging of clusters. This causes a pronounced increase in sequence dropout for these codec-clustering pairings, drastically reducing decoding performance, see Supplementary Table 4.

To highlight this detailed investigation into codec-clustering pairings, we **added a paragraph outlining its findings to the results section** (see ll. 217-230):

“Interestingly, nine of the thirteen tested codecs and code rates performed best with the established general-purpose clustering algorithm CD-HIT (red bars in Fig. 1d). Unsurprisingly, this general preference for CD-HIT was caused by this clusterer’s superior performance, yielding the fewest and least erroneous clusters across all tested clustering algorithms up to an overall error rate of around 10% (see Supplementary Figs. 1-3). In addition, several combinations of clustering algorithms and codecs failed completely (see Supplementary Table 3 for full results). Further investigation suggests incompatibilities exist between these clustering algorithms and the sequence features generated by some codecs (e.g., indexing regions, overlapping sections in the Goldman codec). Specifically, some codec-clustering pairings exacerbate sequence loss by clustering reads from different design sequences together. In some cases, up to 92% of sequences present in the sequencing data no longer appeared in the clustered reads, explaining the incompatibility of such codec-clustering pairings. Thus, the ideal clustering algorithms must reduce the error rate in consensus reads efficiently, thereby reducing the workload for the inner decoder, without causing sequence loss themselves, which drastically burdens codecs’ outer decoders.”

Supplementary Figure 3: Relationship between the number of clusters for each reference sequence and the applied error rate during the basic error scenario for each codec-clustering pairing. For each investigated codec (rows), code rate (columns), and clustering algorithm (color), the number of generated clusters for each reference sequence is shown. As the basic error scenario generates exactly 30 erroneous reads per reference sequence, a low number of clusters per sequence is desirable. However, all clustering algorithms generate multiple clusters for each reference sequence as the applied error rate increases.

Supplementary Table 4: Sequence recovery after clustering 30 perfect reads per reference sequence for each codec-clustering pairing.

Codec	Code rate	Naïve	CD-HIT	Clover	LSH	Starcode
	bit nt ⁻¹	%	%	%	%	%
DNA-Aeon	0.50	100	100	98.2	100	100
	1.00	100	100	98.6	100	100
	1.50	100	100	99.2	100	100
DNA Fountain	0.50	100	99.7	97.8	100	100
	1.00	100	99.9	98.6	100	100
	1.50	100	100	99.1	100	100
DNA-RS	0.50	100	100	98.4	100	100
	1.00	100	100	98.2	100	100
	1.50	100	100	98.2	100	100
Goldman	0.34	100	98.6	79.1	96.3	100
HEDGES	0.63	100	96.5	8.1	99.4	98.0
	1.07	100	99.5	92.4	99.8	99.5
Yin-Yang	1.85	100	100	98.7	100	100

6. Importance of motifs in realistic evaluation: It is also worth mentioning that motifs are highly important for any realistic evaluation. There are known sequence motifs that interact with enzymes during sequencing and also some affecting DNA synthesis. PCR and other processes are also influenced by sequence motifs (or more precisely, by how these motifs interact with enzymes). This important fact should be more prominently highlighted, as only one code in the tested list can handle motifs. This directly relates to my first point: the selection of codes. There are other published codes that can handle sequence motifs. A fair, and more importantly, a realistic comparison would include motif-aware codes. A code that is not able to handle motifs will inevitably fail from time to time in practice, especially when large amounts of data are encoded.

We agree with the reviewer that sequence motifs (together with other sequence properties such as GC content, homopolymer length, and secondary structure) can bias the enzymatic steps involved in DNA data storage. However, based on our review of the literature, quantitative insight into the extent and relevance of sequence motifs for the enzymatic steps is still lacking. Without quantitative data on the effects of sequence motifs however, we consider it difficult to accurately include them in our analysis, without unduly over- or understating the benefits of motif-aware codecs. Nonetheless, we have now **performed a new sensitivity analysis which shows the impact of motif-driven biases in error rate and sequence loss** (see Supplementary Fig. 19), using the motifs implemented in the MESA DNA Simulator by Schwarz et al.¹⁶

We find that motif-driven biases in both error rate and sequence loss can have considerable impact on decoding performance, if they are sufficiently strong (i.e., the likelihood of errors in or sequence loss

of a motif-containing sequence is increased by >10x). We have **highlighted our analysis on the effects of sequence motifs to the discussion** (see ll. 546-549):

“For example, only the DNA-Aeon codec can be constrained to avoid biological motifs (see Table 1), which can negatively affect codec performance if motif-containing sequences are considerably more prone to errors or dropout (see Supplementary Fig. 19).”

Supplementary Figure 19: Benefits of motif avoidance during error-correction coding. The DNA-Aeon codec supports custom codebooks to avoid the introduction of user-defined motifs into the design sequences. To test the potential benefit of motif avoidance, a baseline scenario of 3% errors (53% substitutions, 45% deletions, 2% insertions) and 1% sequence dropout was modified such that all sequences containing any of the 46 motifs with biological meaning defined in the MESA DNA Simulator by Schwarz et al.¹⁶ either had drastically increased error rates (**a**) or were drastically more likely to drop out of the sequencing data (**b**). Using a custom codebook avoiding these motifs with DNA-Aeon (right-most bar) precluded any negative effect of increased error rates or sequence loss, as expected.

In conclusion, the paper provides a valuable and ambitious framework for standardized code benchmarking in DNA data storage. However, its conclusions are somewhat constrained by the limited code selection, oversimplified error models, lack of parameter optimization, and absence of computational performance metrics. Expanding the scope and granularity of the evaluation would further enhance the robustness and applicability of the findings.

We thank the reviewer for highlighting the value of our study and believe our changes have sufficiently expanded the scope and granularity of our benchmarking. We believe our efforts to expand the analysis of codec runtime and introduce an investigation into codec-clustering pairings considerably improved the applicability of our findings.

References

1. Zan, X., Xie, R., Yao, X., Xu, P. & Liu, W. A Robust and Efficient DNA Storage Architecture Based on Modulation Encoding and Decoding. *J. Chem. Inf. Model.* **63**, 3967–3976 (2023).
2. Su, Y. *et al.* A Robust and Efficient Representation-based DNA Storage Architecture by Deep Learning. *Small Methods* **9**, 2400959 (2025).
3. Zheng, X. *et al.* A generative adversarial network for multiple reads reconstruction in DNA storage. *Sci. Rep.* **14**, 32071 (2024).
4. Chandak, S. *et al.* Improved read/write cost tradeoff in DNA-based data storage using LDPC codes. in *2019 57th Annual Allerton Conference on Communication, Control, and Computing* 147–156 (2019).
5. Song, L. *et al.* Robust data storage in DNA by de Bruijn graph-based de novo strand assembly. *Nat. Commun.* **13**, 1–9 (2022).
6. Welzel, M. *et al.* DNA-Aeon provides flexible arithmetic coding for constraint adherence and error correction in DNA storage. *Nat. Commun.* **14**, 628 (2023).
7. Antkowiak, P. L. *et al.* Low cost DNA data storage using photolithographic synthesis and advanced information reconstruction and error correction. *Nat. Commun.* **11**, 5345 (2020).
8. Meiser, L. C. *et al.* Reading and writing digital data in DNA. *Nat. Protoc.* **15**, 86–101 (2019).
9. Erlich, Y. & Zielinski, D. DNA Fountain enables a robust and efficient storage architecture. *Science* **355**, 950–954 (2017).
10. Zhao, X. *et al.* Composite Hedges Nanopores codec system for rapid and portable DNA data readout with high INDEL-Correction. *Nat. Commun.* **15**, 9395 (2024).
11. Ping, Z. *et al.* Towards practical and robust DNA-based data archiving using the yin–yang codec system. *Nat. Comput. Sci.* **2**, 234–242 (2022).
12. Gimpel, A. L., Stark, W. J., Heckel, R. & Grass, R. N. A digital twin for DNA data storage based on comprehensive quantification of errors and biases. *Nat. Commun.* **14**, 6026 (2023).
13. Heckel, R., Mikutis, G. & Grass, R. N. A Characterization of the DNA Data Storage Channel. *Sci. Rep.* **9**, 1–12 (2019).
14. Chen, Y.-J. *et al.* Quantifying molecular bias in DNA data storage. *Nat. Commun.* **11**, 1–9 (2020).
15. Ceze, L., Nivala, J. & Strauss, K. Molecular digital data storage using DNA. *Nat. Rev. Genet.* **2019** **20**, 456–466 (2019).
16. Schwarz, M. *et al.* MESA: automated assessment of synthetic DNA fragments and simulation of DNA synthesis, storage, sequencing and PCR errors. *Bioinformatics* **36**, 3322–3326 (2020).

Response to Referees

Reviewer comments in *italics*, author replies in **red** with actions **bolded**.

Line numbers refer to the revised manuscript, in which changes have been highlighted in **yellow**.

Comments by Referee #1

The authors have solved most of my concerns, but I still concern the following issues:

1. I insist that the authors should select real SOTA codec to make comparison in the main text instead of adding some results in supplementary section.

We follow the reviewer's suggestion to move the comparison of other codecs to the main text. We have therefore **moved Supplementary Fig. 21 to the main manuscript, as the new Fig. 5**. In addition, we have considerably **extended our discussion of this comparison to other codecs in the results section (ll. 479-509)**:

“Finally, to highlight the utility of the developed benchmarking pipeline for codec comparisons, the scope of literature codecs was extended to cover additional codecs by Song et al. (DBGPS)¹, Chandak et al. (LDPC)² and Zan et al. (Modulation)³. Figure 5 shows the results of the *in silico* benchmarks for these three additional codecs, with the results by the best-performing codecs from the main study – DNA-Aeon by Welzel et al.⁴ and DNA-RS by Heckel et al.^{5,6} – also shown for comparison. In line with the results from Fig. 1d, clustering by CD-HIT drastically improved error tolerance across these additional codecs, see Fig. 5a. Moreover, these codecs' ability to correct individual error types did not differ from the results of DNA-Aeon and DNA-RS (see Fig. 5b).

Importantly, the Pareto fronts across code rates in Fig. 5c+d show that DBGPS by Song et al.¹ performed similarly to DNA-Aeon and DNA-RS in most cases. Thus, its performance is on-par with these state-of-the-art codecs, to within the experimental error associated with the lack of parameter optimization in this study. In contrast, both LDPC by Chandak et al.² and the modulation-based codec by Zan et al.³ fall behind the performance of DNA-Aeon, DNA-RS, and DBGPS in our *in silico* benchmarks.

Taken together, the extension of our *in silico* pipeline to three additional codecs from the literature demonstrates its simplicity and utility for codec benchmarking, without requiring any experimental effort. In addition, the results obtained for the DBGPS codec by Song et al.¹ further support the observation that a well-executed, balanced inner/outer code separation strategy is more important for codec performance than the specific choice of employed error-correction code. Thus, these results showcase both how error-correction codecs for DNA data storage are optimally constructed and how their performance is realistically assessed.”

2. I suspect the result in Supplementary Figure 21 (a) and (b) about the error tolerance rate about modulation based codec as it has a supper error correction capability than any current codecs.

In our analysis, under the expected error levels given, the modulation-based codec by Zan et al.³ performs worse than other codecs from the literature from our selection. We note that the difference in performance between our study and the study by Zan et al.³ is likely caused by two reasons:

1. The modulation-based codec by Zan et al.³ does not implement any redundancy for error-correction on the data level (such as the Reed-Solomon or Fountain codes by other codecs, see Table 1). Instead, it relies fully on the error-correcting capability of clustering and sequence alignment. As a result, it fails to decode the data if any design sequence is missing or contains any error after consensus generation.
2. The codec implementation provided by Zan et al.³ on GitHub does not perform sequence indexing. Thus, decoding also fails if the decoder receives sequences out of order.

As a result, the codec is missing from the Pareto plots in Fig. 5c+d, which show benchmarks including both sequence dropout and unordered output. **We now include a detailed discussion of differences in performance assessments between codecs and their limitations in the results section (II. 494-502):**

“In the case of the modulation-based codec by Zan et al.³, its lack of data-level error correction renders it comparable to the Yin-Yang codec by Ping et al.⁷ investigated in our initial codec benchmark. Reassuringly, both of these codecs relying solely on clustering and sequence alignment for error correction achieved similar tolerated error rates of around 4-5% in our test (compare Fig. 1d and 5a). However, contrary to the Yin-Yang codec by Ping et al.⁷, the implementation of the modulation-based codec provided in the repository by Zan et al.³ does not include sequence indexing. As a result, it cannot recover the stored data if sequences are recovered out of order, unless sequence indexing is implemented separately from the codec. These two limitations of the modulation-based codec cause the discrepancy in observed performance between Zan et al.³ and the benchmark in Fig. 5.”

3. In fig 1(d), concerning the error rate per nt, I want to know what the error types is, it means the sum of the insertion-deletion-substitution? further, what the ratio of the three errors? this is very important as I mention last time that in DNA storage, the unique error pattern is the complex IDS errors which makes the information retrieval is very challenge.

The reviewer’s interpretation of the error rate in Fig. 1d is correct. The error rate in this benchmark is the sum of the insertion, deletion, and substitution rates. Their ratio is fixed at 53% substitutions, 45% deletions, and 2% insertions. This resembles the error pattern we observed experimentally in Ref. 8, and is similar to the error pattern we observed experimentally in this study (see Suppl. Fig. 15a+b). We have **added this important information to the figure legend (II. 211-213):**

“The error rate shown corresponds to the combined rate of substitutions, deletions, and insertions at a fixed composition of 53% substitutions, 45% deletions, and 2% insertions, thereby resembling the experimental error pattern in Ref. 8.”

4. As this paper intend to make a comprehensive comparison of SOTA codes in error correction, it is very valuable to discuss which codec have the potential to deal with future large scale applications considering cost, time and error capability.

The reviewer raises an important point regarding the suitability of the different codecs for large-scale applications in the future. DNA data storage is still in its infancy, but one can observe that coding principles previously developed for other problems perform well for DNA data storage, if appropriately adapted. We have now **included additional discussion on this observation, together with the remaining open problems of insertion/deletion correcting codes and broken sequence reconstruction. (II. 580ff):**

"Our analysis shows that codec comparison and benchmarking is important for discussing new coding schemes, and our *in silico* pipeline contributes suitable benchmarking tools to the current literature. In addition, our analysis demonstrates that although DNA data storage represents an error channel with error patterns different to preceding data storage technologies, error-correcting principles developed for existing channels remain highly relevant. Consequently, the state-of-the-art codes for DNA data storage leveraging existing Reed-Solomon-, Fountain- and LDPC-codes perform best if in-sequence error correction is combined with approaches to reconstruct lost sequences and compensate for remaining errors within the sequences. An additional advantage of using pre-existing coding principles is the certainty that such codes can be brought to production level coding/decoding speeds with appropriate implementation into dedicated hardware, as e.g. in CD-ROM drives.⁹ Under the DNA data storage scenarios investigated in this work, constrained coding approaches do not have competitive edge over non-constrained coding, as also discussed in detail by Weindel et al.¹⁰ However, better codes for correcting deletions and insertions might be beneficial for future DNA storage systems and allow data reconstruction in error prone scenarios even without the current aligning of multiple reads for insertion/deletion correction."

References

1. Song, L. *et al.* Robust data storage in DNA by de Bruijn graph-based de novo strand assembly. *Nat. Commun.* **13**, 1–9 (2022).
2. Chandak, S. *et al.* Improved read/write cost tradeoff in DNA-based data storage using LDPC codes. in *2019 57th Annual Allerton Conference on Communication, Control, and Computing* 147–156 (2019).
3. Zan, X., Xie, R., Yao, X., Xu, P. & Liu, W. A Robust and Efficient DNA Storage Architecture Based on Modulation Encoding and Decoding. *J. Chem. Inf. Model.* **63**, 3967–3976 (2023).
4. Welzel, M. *et al.* DNA-Aeon provides flexible arithmetic coding for constraint adherence and error correction in DNA storage. *Nat. Commun.* **14**, 628 (2023).
5. Antkowiak, P. L. *et al.* Low cost DNA data storage using photolithographic synthesis and advanced information reconstruction and error correction. *Nat. Commun.* **11**, 5345 (2020).
6. Meiser, L. C. *et al.* Reading and writing digital data in DNA. *Nat. Protoc.* **15**, 86–101 (2019).
7. Ping, Z. *et al.* Towards practical and robust DNA-based data archiving using the yin–yang codec system. *Nat. Comput. Sci.* **2**, 234–242 (2022).
8. Gimpel, A. L., Stark, W. J., Heckel, R. & Grass, R. N. A digital twin for DNA data storage based on comprehensive quantification of errors and biases. *Nat. Commun.* **14**, 6026 (2023).
9. Geisel, W. A. *Tutorial on Reed-Solomon Error Correction Coding*. <https://ntrs.nasa.gov/citations/19900019023> (1990).
10. Weindel, F., Gimpel, A. L., Grass, R. N. & Heckel, R. Embracing Errors Can Be More Efficient Than Avoiding Them Through Constrained Coding for DNA Data Storage. *IEEE Trans. Mol. Biol. Multi-Scale Commun.* 1–1 (2025) doi:10.1109/TMBMC.2025.3610330.

Response to Referees

Reviewer comments in *italics*, author replies in **red** with actions **bolded**.

Line numbers refer to the revised manuscript, in which changes have been highlighted in **yellow**.

Comments by Referee #1

The authors have solved part of my concerns, but I still have some main concerns:

1. The YY code has no error correction ability, but your result shows it tolerates about 4% errors (Fig 1d). Could you explain the reason? Moreover, why did you select it as a SOTA method to study the error correction capability in this paper?

The Yin-Yang codec by Ping et al.¹ does indeed not have error-correction capabilities itself. As already described in ll. 187-190 of the manuscript, this codec nevertheless tolerates around 4% of errors in our basic error scenario because of the error-correcting effect of clustering and consensus generation. Up to an error rate of 4%, the consensus reads after clustering do not contain any errors and thus no codec-level error correction is required to recover the information.

We have extended the existing explanation to highlight how the Yin-Yang codec is able to recover the data at this error rate (ll. 190-192):

“Specifically, the clustering step already completely eliminates all errors in the consensus reads up to this error rate (see also Supplementary Fig. 1), removing the need for any codec-level error correction for error-free data recovery.”

We note that the Yin-Yang codec by Ping et al.¹ was not selected as a SOTA method. It was included for benchmarking in this study, but only the best-performing codecs (DNA-Aeon, DNA-RS, and DBGPS) were selected as the state-of-the-art in the manuscript. Perhaps the inclusion of the Yin-Yang codec in Table 1 of the manuscript was misleading. However, this table does not list the codecs which we identify as the state-of-the-art. Rather, it lists the codecs which were included in the main benchmarking study.

To make the distinction between codecs included in the study and those selected as the state-of-the-art clearer, we have adjusted the caption of Table 1 (ll. 102-105):

“Table 1: Codecs included in the in silico and in vitro benchmarks, as well as their properties. Clustering denotes the clustering approach used in the original study, if any. The pairing of codecs with clustering algorithms used in this study is detailed in Supplementary Table 3. Based on the benchmarking results (see below), only the DNA-Aeon and DNA-RS codecs were selected as the current state-of-the-art.”

2. AI-based techniques have been introduced to solve the error issues in DNA storage, and recent studies showed that they have promising results both in error correction, data compression and encoding etc. But this paper neglects the function of these studies. As I know, compared with the traditional methods used in this paper, deep learning methods are more suitable for future DNA storage.

We agree with the reviewer that machine learning approaches are actively being investigated for DNA data storage. Unfortunately, we were unable to include such codecs despite our best efforts because the corresponding codes and models are not available.

Specifically, the studies on deep learning-based methods by Su et al.² and Zheng et al.³ did not deposit the corresponding code, rendering it impossible for us to include these codecs in our study. While the study by Bar-Lev et al.⁴ did deposit the corresponding code, we were unable to use their code due to poor documentation and multiple implementation challenges. Specifically, the code contains many hard-coded paths and requires multiple compilations of C++ code during the encoding and decoding steps. This lack of code portability rendered implementation of this codec impossible on our computing cluster. Finally, retraining of the deep learning models would be required to obtain representative performance, which would be outside the scope of our study.

To facilitate this study, we had to define criteria to determine which codes to include into the study, as described in ll. 111-115. These criteria were:

- The full open-source implementation of the code is available. While we have performed tweaking of software implementations (see Supplementary Note 1), it was outside the scope of this comparative study to re-implement or adapt codecs, for which not code is available.
- The corresponding study includes in vitro experiments to demonstrate suitability for realistic experimental workflows.
- The corresponding study was published prior to the cut-off date of October 2023 (when we started the study). As our study includes vast testing of software as well as in vitro experiments, which took considerable time.

Apart of not being reproducible by us, the study by Bar-Lev et al.⁴ also does not fulfil the last criterium.

Thus, **we added a paragraph to the discussion to note the ongoing efforts to improve codecs using deep learning and to highlight how our benchmarking approach can help with ensuring code reproducibility** (ll. 605-608):

“For example, recent efforts to employ deep learning in DNA data storage²⁻⁴ were not covered in this study, but could benefit from the benchmarking scheme proposed in this study to comprehensively assess their scalability and transferability.”

3. Concerning the modulation-based method, I agree it could not solve sequence loss, but HEDGES and DBGPS also can not deal with such issues. I still suspect the reported results in this paper.

While not a key focus of these studies, the implementations of both the HEDGES codec by Press et al.⁵ and the DBGPS codec by Song et al.⁶ can tolerate and correct sequence loss, contrary to the reviewer’s statement. This is shown both by the results in the manuscript (see Fig. 1c for HEDGES and Fig. 5c for DBGPS) and noted in the original studies of these codecs:

- In Press et al.⁵, the authors state: “we used an outer-code concatenated design with packets of 255 strands of length 300 protected by RS(255,223)”⁵ and “erasures (unknown bits or bytes) [...] can be corrected in the outer code”⁵. The authors also state that the deposited code implements an outer RS code: “The computer code used for the generation and testing of the inner HEDGES code and outer RS code is available at <https://github.com/whpress/hedges>.”⁵
- In Song et al.⁶, the authors state: “The strand redundancy was set to 7.8%, which supports reliable data recovery when the decoder receives more than 95% of strands”⁶ and “raw reads

generated were processed by DBGPS for strand reconstruction followed by decoding using the outer fountain codes”⁶. The authors also state that the deposited code implements an outer Fountain code: “Python implementation of DBGPS using fountain codes as outer codes is available at <https://doi.org/10.5281/zenodo.6833784>”⁶ (see SI)

4. Last, the loss issue in DNA storage is usually solved by fountain code or inner-outer code strategy. Therefore, base errors in sequences are usually corrected by various methods, which may combine the fountain code or inner-outer code strategy.

We agree with the reviewer that sequence loss in DNA data storage is commonly addressed using fountain codes and an inner-outer code architecture, as noted in the manuscript (e.g., ll. 55-57). However, it is important to emphasize that implementing these mechanisms to mitigate sequence loss is a fundamental requirement for any practical DNA data storage codec. For this reason, we believe it is valuable to point out that some of the evaluated codec implementations lack this crucial capability, as doing so underscores the significance of sequence loss and provides important information for users.

We have added an explanation about the potential retrofitting of codecs with an outer code, and our rationale against considering this in our benchmarks, to the manuscript in ll. 505-509:

“Naturally, features such as sequence indexing and an outer code for tolerance to sequence loss can be retrofitted to existing codecs, using established methods like Fountain or Reed-Solomon codes. However, such adaptations to existing codec implementations require additional programming, testing and optimization efforts, which is not realistic for most use cases and was therefore not in scope of this study.”

References

1. Ping, Z. *et al.* Towards practical and robust DNA-based data archiving using the yin–yang codec system. *Nat. Comput. Sci.* **2**, 234–242 (2022).
2. Su, Y. *et al.* A Robust and Efficient Representation-based DNA Storage Architecture by Deep Learning. *Small Methods* **9**, 2400959 (2025).
3. Zheng, X. *et al.* A generative adversarial network for multiple reads reconstruction in DNA storage. *Sci. Rep.* **14**, 32071 (2024).
4. Bar-Lev, D., Orr, I., Sabary, O., Etzion, T. & Yaakobi, E. Scalable and robust DNA-based storage via coding theory and deep learning. *Nat. Mach. Intell.* **7**, 639–649 (2025).
5. Press, W. H., Hawkins, J. A., Jones, S. K., Schaub, J. M. & Finkelstein, I. J. HEDGES error-correcting code for DNA storage corrects indels and allows sequence constraints. *Proc. Natl. Acad. Sci.* **117**, 18489–18496 (2020).
6. Song, L. *et al.* Robust data storage in DNA by de Bruijn graph-based de novo strand assembly. *Nat. Commun.* **13**, 1–9 (2022).